# SHARP LEARNING BOUNDS FOR CONTRASTIVE UNSUPERVISED REPRESENTATION LEARNING

## ABSTRACT

Contrastive unsupervised representation learning (CURL) encourages data representation to make semantically similar pairs closer than randomly drawn negative samples, which has been successful in various domains such as vision, language, and graphs. Although recent theoretical studies have attempted to explain its success by upper bounds of a downstream classification loss by the contrastive loss, they are still not sharp enough to explain an experimental fact: larger negative samples improve the classification performance. This study establishes a downstream classification loss bound with a tight intercept in the negative sample size. By regarding the contrastive loss as a downstream loss estimator, our theory not only improves the existing learning bounds substantially but also explains why downstream classification empirically improves with larger negative samples—because the estimation gap of the downstream loss decays with larger negative samples. We verify that our theory is consistent with experiments on synthetic, vision, and language datasets.

## 1 INTRODUCTION

The contrastive loss (Chopra et al., 2005) is one of the popular loss functions in metric learning (Kulis, 2012) and representation learning (Bengio et al., 2013). The contrastive loss forces data representation of semantically similar pairs closer in some metric space than multiple random samples, called negative samples. Many state-of-the-art representation learning algorithms use a type of contrastive losses in natural language processing (Mikolov et al., 2013; Logeswaran & Lee, 2018), vision (Chopra et al., 2005; He et al., 2019; Chen et al., 2020), and graph (Lirong et al., 2021) domains. A simple model built on top of the learned representation can achieve almost the same accuracy as supervised learning does. See surveys (Le-Khac et al., 2020; Schmarje et al., 2021) and references therein for recent empirical progress.

Compared to its empirical success, we still do not know much about the theoretical perspective of the contrastive loss function. Since the first theoretical analysis of contrastive unsupervised representation learning (CURL) (Arora et al., 2019), a number of follow-up researches (Nozawa et al., 2020; Chuang et al., 2020; Nozawa & Sato, 2021; Ash et al., 2021) improved theoretical understanding of CURL. They provided upper bounds of a downstream classification loss by the contrastive loss. While they partially explain why CURL provides good representation in terms of downstream classification, a controversy has been held on the impact of the negative sample size, denoted by $K$ in this work. From the empirical viewpoint, it has been widely shown that downstream classification accuracy tends to be better with the larger $K$ (He et al., 2020; Chen et al., 2020).[1] From the theoretical viewpoint, although Ash et al. (2021) and Nozawa & Sato (2021) addressed the issue of the earlier analysis (Arora et al., 2019)—the bound grows exponentially in $K$—their improved bounds are still so huge that they may not capture the actual behavior of CURL related to $K$, as we will see in Figure 1. In addition, Ash et al. (2021) claims that the optimal $K$ exists in smaller $K$, which does not agree well with the empirical observations. Henceforth, the relationship between the performance of downstream classification and $K$ needs to be investigated carefully.

This study aims to clarify the relationship between the performance of downstream classification and the negative sample size $K$ by deriving an upper bound of the downstream classification loss

---

[1]The empirical evidence has also not reached a consensus yet. Chen et al. (2021) reported that SimCLR with the smaller $K$ could be competitive in linear evaluation than $K$ used by the original SimCLR (Chen et al., 2020).

(Theorem 1). By deriving the *lower* bound (Theorem 2), we found that the intercept of our upper bound is tight in $K$. Notably, such a lower bound has yet to be known in CURL so far. Our bounds support that the larger $K$ reduces estimation gap of the downstream classification loss (Section 3.2), while the smaller $K$ could perform well. This finding is evidence for the empirical success of the larger $K$. In addition, our bounds are compared with the existing bounds to demonstrate that there is no optimal point in $K$ and how accurately the derived bounds capture the downstream loss (Section 4). Finally, we empirically verify our theory by experiments (Section 6) on a synthetic dataset, CIFAR-10/100 (Krizhevsky, 2009) datasets, and Wiki-3029 dataset (Arora et al., 2019).

## 2 FORMULATION OF CONTRASTIVE LEARNING

First, this section briefly summarizes the problem setup and formulation of CURL.

**Notation.** The $C$-dimensional vector whose elements are all one is denoted by $\mathbf{1}_C := [1 \ 1 \ \dots \ 1]^\top$. When it is clear from context, the subscript is abbreviated. For a vector $\mathbf{a} \in \mathbb{R}^p$, $a_{(i)}$ denotes the $i$-th largest element of $\mathbf{a}$, namely, $a_{(1)} \geq a_{(2)} \geq \dots \geq a_{(p)}$. Likewise, $a_{(-i)}$ denotes the $i$-th smallest element of the vector $\mathbf{a}$. The indicator function is denoted by $\mathbb{1}_{\{A\}}$ for a predicate $A$. Let $\triangle^C := \{\mathbf{p} \in [0,1]^C \mid \mathbf{p}^\top \mathbf{1} = 1\}$ be the $C$-dimensional probability simplex. For $\mathbf{p} \in \triangle^C$, the Shannon entropy is denoted by $\mathbb{H}(\mathbf{p}) := -\sum_{c \in [C]} p_c \ln p_c$. Let $H_n$ be the $n$-th harmonic number.

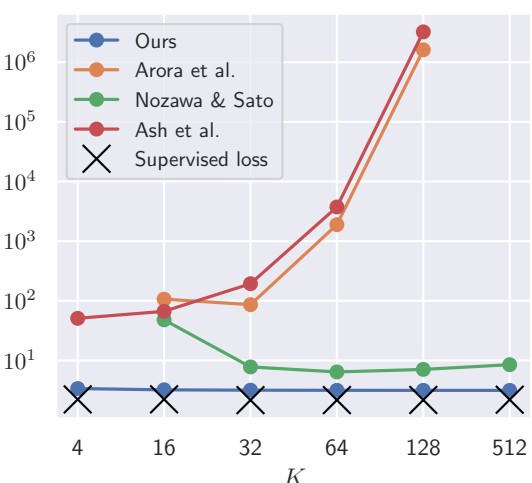

Figure 1: Empirical comparison of our upper bound and the existing bounds on CIFAR-10 ($C = 10$). Arora et al.'s and Nozawa & Sato's bounds are valid only at $K + 1 \geq C$. Note that Arora et al.'s and Ash et al.'s bounds become infinity at $K = 512$. As can be seen, our bound is the closest approximator of the true mean supervised losses. The detailed experimental setup is stated in Section 6.2.

**Supervised classification.** One of the goals in machine learning is supervised classification, while we consider the setup where supervision is unavailable. Here, we first formulate the multi-class classification problem. Assume that there exists an unknown number of latent classes denoted by $C \in \mathbb{N}$. Let $\mathcal{X}$ be $d$-dimensional feature space and $\mathcal{Y} := [C]$ be the supervised label set.[2] In the supervised setup, we are interested in the following risk quantity, the *supervised loss*, for a multi-class classifier $\mathbf{g} : \mathcal{X} \to \mathbb{R}^C$:

$$R_{\text{supv}}(\mathbf{g}) := \mathop{\mathbb{E}}_{\mathbf{x}, y \sim \mathbb{P}} \left[ -\ln \frac{\exp(g_y(\mathbf{x}))}{\sum_{c \in \mathcal{Y}} \exp(g_c(\mathbf{x}))} \right], \tag{1}$$

which is specialized for the softmax cross-entropy loss. The expectation is taken over the unknown underlying joint distribution $\mathbb{P}$. The classifier is eventually used for prediction by $\text{argmax}_{y \in \mathcal{Y}} g_y(\cdot)$.

**Contrastive unsupervised representation learning (CURL).** In the CURL framework (Arora et al., 2019), we target to learn meaningful data representation in an unsupervised manner. By doing so, a similarity model is trained to make the representation of *positive* pairs more similar than randomly drawn $K$ *negative* samples using a contrastive loss. The class-conditional distribution is denoted by $\mathcal{D}_c := \mathbb{P}(X \mid Y = c)$ for each $c \in \mathcal{Y}$ and the class-prior distribution by $\boldsymbol{\pi} := [\mathbb{P}(Y = c)]_{c \in \mathcal{Y}} \in \triangle^C$. The data generating process is described as follows: (i) draw latent positive/negative classes: $c^+$,

---

[2]Strictly speaking, latent classes should be distinguished from the supervised labels as they may contain concepts that human annotators are not interested in. In the CURL framework (Arora et al., 2019), latent classes $[C]$ and the supervised label set $\mathcal{Y}$ are allowed to be different, i.e., $\mathcal{Y} \subseteq [C]$. To focus on the relationship between the supervised loss and contrastive loss, we analyze under the assumption $\mathcal{Y} = [C]$ for simplicity.

$\{c_k^-\}_{k=1}^K \sim \boldsymbol{\pi}^{K+1}$ (ii) draw an anchor sample $\mathbf{x} \sim \mathcal{D}_{c^+}$ (iii) draw a positive sample $\mathbf{x}^+ \sim \mathcal{D}_{c^+}$[3] (iv) draw $K$ negative samples $\mathbf{x}_k^- \sim \mathcal{D}_{c_k^-}$ (for each $k \in [K]$).

Now we specify our model. A multi-class classifier $\mathbf{g} : \mathcal{X} \to \mathbb{R}^C$ consists of representation $\mathbf{f} : \mathcal{X} \to \mathbb{R}^h$ and linear parameters $\mathbf{W} \in \mathbb{R}^{C \times h}$ as $\mathbf{g}(\cdot) := \mathbf{W}\mathbf{f}(\cdot)$, where $h \in \mathbb{N}$ denotes the dimensionality of the representation given in advance. In CURL, the representation is learned through minimization of the following *contrastive loss*

$$R_{\mathrm{cont}}(\mathbf{f}) := \mathop{\mathbb{E}}_{\substack{c^+, \{c_k^-\}_{k=1}^K \\ \sim \boldsymbol{\pi}^{K+1}}} \mathop{\mathbb{E}}_{\substack{\mathbf{x}, \mathbf{x}^+ \sim \mathcal{D}_{c^+}^2 \\ \mathbf{x}_k^- \sim \mathcal{D}_{c_k^-}}} \left[ -\ln \frac{\exp(\mathbf{f}(\mathbf{x})^\top \mathbf{f}(\mathbf{x}^+))}{\exp(\mathbf{f}(\mathbf{x})^\top \mathbf{f}(\mathbf{x}^+)) + \sum_{k \in [K]} \exp(\mathbf{f}(\mathbf{x})^\top \mathbf{f}(\mathbf{x}_k^-))} \right]. \quad (2)$$

In the downstream task, the representation $\mathbf{f}$ shall be frozen, and the parameters $\mathbf{W}$ are to be learned via minimizing the supervised loss $R_{\mathrm{supv}}$.

**Evaluation of representations.** For the sake of evaluation, a specific linear classifier called *mean classifier* is introduced. Given representation $\mathbf{f}$, the mean classifier $\mathbf{W}^\mu$ is defined as $\mathbf{W}^\mu := [\boldsymbol{\mu}_1 \cdots, \boldsymbol{\mu}_C]^\top$, where $\boldsymbol{\mu}_c := \mathbb{E}_{\mathbf{x} \sim \mathcal{D}_c}[\mathbf{f}(\mathbf{x})]$. This will later be used for evaluating the representation $\mathbf{f}$ combined with the supervised loss, which is denoted by $R_{\mu\text{-supv}}(\mathbf{f}) := R_{\mathrm{supv}}(\mathbf{W}^\mu \mathbf{f})$. We call it the *mean supervised loss*. If the mean supervised loss is successfully bounded from above, we end up a bound on the supervised loss through $\inf_{\mathbf{W} \in \mathbb{R}^{C \times h}} R_{\mathrm{supv}}(\mathbf{W}\mathbf{f}) \le R_{\mu\text{-supv}}(\mathbf{f})$. For this reason, an upper bound on $R_{\mu\text{-supv}}$ is an intermediate milestone that we seek throughout this paper.

## 3 LEARNING BOUNDS FOR CONTRASTIVE LEARNING

In this section, our main theoretical results are provided. We aim at showing that the contrastive loss $R_{\mathrm{cont}}(\mathbf{f})$—the computable functional in CURL—serves as a good estimator of the mean supervised loss $R_{\mu\text{-supv}}(\mathbf{f})$—the inaccessible target in CURL—for any $\mathbf{f}$. We show this by establishing upper and lower bounds of $R_{\mu\text{-supv}}(\mathbf{f})$ by $R_{\mathrm{cont}}(\mathbf{f})$. Eventually, the minimization of $R_{\mathrm{cont}}(\mathbf{f})$ may lead to a good minimizer of $R_{\mu\text{-supv}}(\mathbf{f})$. All proofs are provided in Appendix B.

### 3.1 MAIN RESULTS

First, we show a sharp upper bound of the mean supervised loss. Unlike any of the existing learning bounds of CURL, the upper bound obtained here has a constant coefficient in the contrastive loss and is applicable for all $C$ and $K$ (see discussions in Section 4).[4]

**Theorem 1.** *For all $\mathbf{f}$ such that $\|\mathbf{f}(\mathbf{x})\|_2 \le L$ ($\forall \mathbf{x} \in \mathcal{X}$), the following inequality holds.*

$$R_{\mu\text{-supv}}(\mathbf{f}) \le R_{\mathrm{cont}}(\mathbf{f}) + \Delta_{\mathrm{U}}, \quad (3)$$

*where $\Delta_{\mathrm{U}} := 2\ln((CK\pi_{(-1)} + 1)\cosh(L^2)) - 2\ln(1 + K) - \ln K\pi_{(-1)}$.*

Next, the first lower bound of the mean supervised loss is provided. While the existing theoretical analyses only provided upper bounds, they often entail a huge coefficient in the contrastive loss. The lower bound provided below has the same constant coefficient and intercept ($\Delta_{\mathrm{U}}$ and $\Delta_{\mathrm{L}}$) rate as our upper bound, ensuring the tightness of our analysis.

**Theorem 2.** *For all $\mathbf{f}$ such that $\|\mathbf{f}(\mathbf{x})\|_2 \le L$ ($\forall \mathbf{x} \in \mathcal{X}$), the following inequality holds.*

$$R_{\mu\text{-supv}}(\mathbf{f}) \ge R_{\mathrm{cont}}(\mathbf{f}) + \Delta_{\mathrm{L}}, \quad (4)$$

*where $\Delta_{\mathrm{L}} := \mathbb{H}(\boldsymbol{\pi}) + \ln \frac{K}{(K+1)^2} - 2\ln\cosh(L^2)$.*

---

[3]In self-supervised learning, data augmentation (DA) is commonly used to obtain a positive sample. While Nozawa & Sato (2021) handled DA in this formulation, we omit it for simplicity to focus on the effect of $K$.

[4]Even if $\mathcal{Y} = [C]$ is assumed in Section 2, Theorem 1 can be extended to the case $\mathcal{Y} \subseteq [C]$ (subset) and $\mathcal{Y}$ is a coarse-grained set of $[C]$. In contrast, Theorem 2 can only be extended to the coarse-grained $\mathcal{Y}$. The details are discussed in their proofs.

Our proofs leverage that the contrastive loss and mean supervised loss share the similar log-sum-exp functional form, which leads to the direct relationships between the two losses. This is in contrast to the existing works including Arora et al. (2019), which approximate the mean supervised loss with the contrastive loss by taking the expectation over latent classes, leading to an exponentially large coefficient.

In Theorems 1 and 2, the size of the representation $\|\mathbf{f}(\mathbf{x})\|_2$ is assumed to be bounded. This assumption is reasonable from the experimental perspective since it is common to normalize representation to employ the cosine similarity as the similarity metric. Several works reported that the normalized embeddings improve the performance (Chen et al., 2020; Wang & Isola, 2020). Unlike the existing analyses (reviewed in Section 4), we take advantage of this assumption to derive the sharp bounds.

As we see in Section 3.2, $\Delta_{\mathrm{U}}$ and $\Delta_{\mathrm{L}}$ are the same order in $K$ under the uniform class prior assumption. By applying either the high-probability bound (Arora et al., 2019) or PAC-Bayesian analysis (Nozawa et al., 2020), Theorem 1 (Theorem 2 as well) can be naturally extended to the form $R_{\mu\text{-supv}}(\widehat{\mathbf{f}}) \leq R_{\mathrm{cont}}(\mathbf{f}) + \Delta_{\mathrm{U}} + \chi$ with a complexity term $\chi$, where $\widehat{\mathbf{f}}$ is the empirical minimizer of the contrastive loss. Since this is a routine, we omit the high-probability bounds.

### 3.2 DISCUSSION

Subsequently, we discuss implications of our main results on the relationship between the mean supervised loss and the negative sample size $K$. For the sake of simplicity, we assume $\pi_c = {}^1/C$ for all $c \in [C]$ (the uniform class prior) in this section.

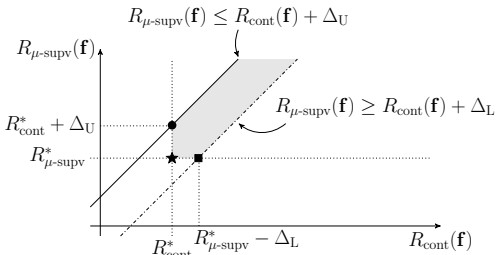

Figure 2: The learning bounds and feasible region. The point ★ , $(R_{\mathrm{cont}}^*, R_{\mu\text{-supv}}^*)$, is the optimal point in the feasible region. The points ● and ■ are mentioned in the texts.

**Gap between learning upper and lower bounds.** Both of our upper (Theorem 1) and lower (Theorem 2) bounds draw the linear relationship between the mean supervised loss $R_{\mu\text{-supv}}$ and the contrastive loss $R_{\mathrm{cont}}$, with the additional intercept terms $\Delta_{\mathrm{U}}$ and $\Delta_{\mathrm{L}}$. Under the uniform class prior assumption, the intercepts are in the same order:

$$\Delta_{\mathrm{U}} = \ln \frac{C}{K} + 2\ln\cosh(L^2) = O(\ln\frac{1}{K}), \quad \Delta_{\mathrm{L}} = \ln \frac{CK}{(K+1)^2} - 2\ln\cosh(L^2) = O(\ln\frac{1}{K}), \quad (5)$$

and the gap between two bounds is $\Delta_{\mathrm{U}} - \Delta_{\mathrm{L}} = 4\ln\cosh(L^2) + 2\ln\left(1 + \frac{1}{K}\right)$, meaning that the gap shrinks to $4\ln\cosh(L^2)$ as $K$ increases. Hence, our bounds have the tight intercepts, and *the larger $K$ is beneficial for CURL from the viewpoint of the estimation gap of the mean supervised loss.*

**Learning bounds and feasible region.** Next, we consider the $(R_{\mathrm{cont}}, R_{\mu\text{-supv}})$-plot, in which a point indicates $(R_{\mathrm{cont}}(\mathbf{f}), R_{\mu\text{-supv}}(\mathbf{f}))$ for some $\mathbf{f}$ (see Figure 2). Here, let us focus on the feasible region in the $(R_{\mathrm{cont}}, R_{\mu\text{-supv}})$-plot by assuming $\|\mathbf{f}\|_2 \leq L$ for any $\mathbf{f}$ (same as Theorems 1 and 2). Then, the mean supervised loss and contrastive loss are essentially lower-bounded by the constants[5]

$$R_{\mu\text{-supv}}^* := \ln\left(1 + (C-1)\exp(-2L^2)\right), \quad (6)$$

$$R_{\mathrm{cont}}^* := \sum_{m=0}^{K} \binom{K}{m}\left(\frac{1}{C}\right)^m \left(1 - \frac{1}{C}\right)^{K-m} \ln\{1 + m + (K-m)\exp(-2L^2)\}, \quad (7)$$

respectively. Hence, the feasible region is

$$R_{\mu\text{-supv}}(\mathbf{f}) \leq R_{\mathrm{cont}}(\mathbf{f}) + \Delta_{\mathrm{U}}, \quad R_{\mu\text{-supv}}(\mathbf{f}) \geq R_{\mathrm{cont}}(\mathbf{f}) + \Delta_{\mathrm{L}}, \quad (8\mathrm{a})$$

$$R_{\mu\text{-supv}}(\mathbf{f}) \geq R_{\mu\text{-supv}}^*, \quad R_{\mathrm{cont}}(\mathbf{f}) \geq R_{\mathrm{cont}}^*, \quad (8\mathrm{b})$$

as illustrated in Figure 2. The first two bounds (8a) restrict the mean supervised loss by the contrastive loss. We specifically refer to these bounds as *learning bounds*. The remaining two bounds (8b)

---

[5]The derivations of $R_{\mu\text{-supv}}^*$ and $R_{\mathrm{cont}}^*$ are detailed in Appendix C.

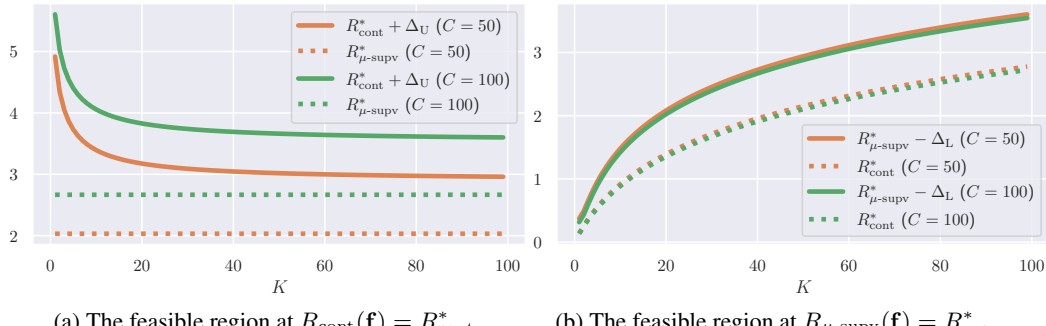

(a) The feasible region at $R_{\text{cont}}(\mathbf{f}) = R^*_{\text{cont}}$.  (b) The feasible region at $R_{\mu\text{-supv}}(\mathbf{f}) = R^*_{\mu\text{-supv}}$.

Figure 3: Visualization of the smallest possible value of $R_{\mu\text{-supv}}$ in the feasible region (8) for different $K$ and $C$. The dotted lines show the essential lower bounds which come from each loss separately. The solid lines show the learning upper bounds in which $R_{\mu\text{-supv}}$ and $R_{\text{cont}}$ restrict each other.

Table 1: Learning bounds of the existing works. Remark that Arora et al.'s and Nozawa & Sato's bounds are valid only $K + 1 \geq C$. The detailed derivations are discussed in Appendix D.

| | UPPER BOUND | REFERENCE |
|---|---|---|
| $R_{\mu\text{-supv}}(\mathbf{f}) \leq$ | $\frac{1}{(1-\tau_K)v_{K+1}}\left\{R_{\text{cont}}(\mathbf{f}) - \mathbb{E}\ln(\text{Col}+1)\right\}$ | Arora et al. (2019) |
| | $\frac{1}{v_{K+1}}\left\{2R_{\text{cont}}(\mathbf{f}) - \mathbb{E}\ln(\text{Col}+1)\right\}$ | Nozawa & Sato (2021) |
| | $\frac{2}{1-\tau_K}\left\lceil\frac{2(C-1)H_{C-1}}{K}\right\rceil\left\{R_{\text{cont}}(\mathbf{f}) - \mathbb{E}\ln(\text{Col}+1)\right\}$ | Ash et al. (2021) |

represent the achievable limits for each loss separately. One of the important questions is how the smallest possible value of $R_{\mu\text{-supv}}$ in the feasible region (8) changes as $K$ and $C$ change. In other words, we are interested in whether the optimal point $(R^*_{\text{cont}}, R^*_{\mu\text{-supv}})$ (★ in Figure 2) is always achievable regardless of the values of $K$ and $C$. To investigate it, we check the following two points.

- The feasible region at $R_{\text{cont}}(\mathbf{f}) = R^*_{\text{cont}}$ (Figure 3a): We plot the value $R^*_{\text{cont}} + \Delta_{\text{U}}$ (solid line; the $R_{\mu\text{-supv}}$-value of the point ● in Figure 2) and the minimum possible $R_{\mu\text{-supv}}$ ($R^*_{\mu\text{-supv}}$; dotted line) numerically. These two curves do not cross for all $K$, which means $R_{\mu\text{-supv}}(\mathbf{f}) = R^*_{\mu\text{-supv}}$ is attainable no matter the values $K$ and $C$. In addition, the bound becomes sharper as $K$ increases, but the gap between the upper bound and $R^*_{\mu\text{-supv}}$ does remain even at the limit $K \nearrow \infty$.

- The feasible region at $R_{\mu\text{-supv}}(\mathbf{f}) = R^*_{\mu\text{-supv}}$ (Figure 3b): When $R_{\mu\text{-supv}}(\mathbf{f}) = R^*_{\mu\text{-supv}}$, the contrastive loss $R_{\text{cont}}(\mathbf{f})$ is upper-bounded by $R^*_{\mu\text{-supv}} - \Delta_{\text{L}}$ (the $R_{\text{cont}}$-value of the point ■ in Figure 2). The curve of this value does not cross $R^*_{\text{cont}}$, which tells us that the lower bound does not exclude the optimal point $(R^*_{\text{cont}}, R^*_{\mu\text{-supv}})$ from the feasible region (8) at any $K$. Note that the gap between $R^*_{\text{cont}}$ and $R^*_{\mu\text{-supv}} - \Delta_{\text{L}}$ gradually increases, meaning that it becomes much easier to attain $R^*_{\mu\text{-supv}}$ as $K$ increases.

Hence, the optimal point ★ stays in the feasible region (8) no matter the value $K$. From this viewpoint, *smaller $K$ is not necessarily disadvantageous because the optimal point ★ remains in the feasible region.* Note again that the estimation of $R_{\mu\text{-supv}}$ may become harder with the smaller $K$ because of the bound gap $\Delta_{\text{U}} - \Delta_{\text{L}} = O(\ln 1/K)$, even if the optimal solution is unaffected by $K$.

**Summary.** We draw a connection between the mean supervised loss and the negative sample size $K$ by the following claim: if we regard CURL as an estimation process of the mean supervised loss by the contrastive loss, the small $K$ does not necessarily have the estimation bias, but the larger $K$ helps reduce the estimation gap.

## 4    COMPARISON WITH EXISTING LEARNING BOUNDS

This section discusses the difference between our main results and the existing theoretical results on CURL. The learning bound analysis of CURL has been first established by Arora et al. (2019) and later extended by Nozawa & Sato (2021) and Ash et al. (2021). Here, we briefly review these results. For ease of the comparison, we assume the uniform class prior, namely, $\pi_c = 1/C$ for all $c \in [C]$. We introduce a notation $\mathrm{Col} := \sum_{k\in[K]} \mathbb{1}_{\{c^+=c_k^-\}}$. Let $v_K$ be the probability that sampled $K$ negative classes contains all classes $c \in [C]$.

$$v_K := \sum_{n=1}^{K} \sum_{m=0}^{C-1} \binom{C-1}{m} (-1)^m \left(1 - \frac{m+1}{C}\right)^{n-1}. \tag{9}$$

The value $v_K$ is often referred to as the *coupon collector's probability*. Let $\tau_K$ be the probability that at least one of the negative classes $c_k^-$ is the same as the positive class $c^+$. Under the uniform class prior, $\tau_K = 1 - (1 - 1/C)^K$. The learning bounds are summarized in Table 1.[6]

**Comparison.** First, the coefficient of $R_{\mathrm{cont}}(\mathbf{f})$ in the upper bounds are numerically compared in Figure 4a. While Theorem 1 has the coefficient 1 for all $K$, the existing bounds have different nature—Arora et al.'s and Ash et al. (2021)'s coefficients have unique minima, and Nozawa & Sato's coefficient has monotonically decreasing nature. Since the larger $K$ was shown advantageous in the several experimental work (Chen et al., 2020; He et al., 2020), the behaviors of Arora et al.'s and Ash et al.'s bounds do not explain it well. In Section 6, we also experimentally confirm the benefits of the large $K$.

Figure 4b compares the upper bound values at $R_{\mathrm{cont}}(\mathbf{f}) = R_{\mathrm{cont}}^*$, namely, the best possible mean supervised loss in terms of the upper bounds. The tendencies are slightly different from Figure 4a—Ash et al.'s bound is monotonically increasing, Arora et al.'s and Nozawa & Sato's bounds have a unique minimum, and ours is monotonically decreasing. Among the compared bounds, only ours agrees well with the experimental fact that the larger $K$ is better.[7]

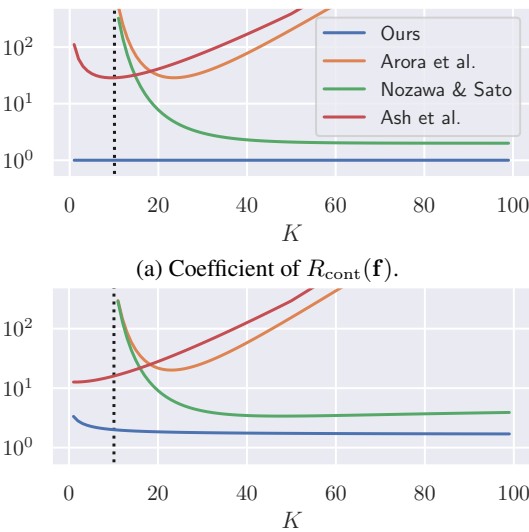

(a) Coefficient of $R_{\mathrm{cont}}(\mathbf{f})$.

(b) Upper bound at $R_{\mathrm{cont}}(\mathbf{f}) = R_{\mathrm{cont}}^*$.

Figure 4: Theoretical comparison of our upper bound and the existing bounds ($C = 10$). Arora et al.'s and Nozawa & Sato's bounds are valid only at $K + 1 \geq C$ (the dotted vertical lines).

Note that the coefficients of the existing theories are so large—the vertical axes are log-scaled—that the upper bounds are vacuous in both situations. In contrast, our theory can bound the downstream classification loss most sharply in all ranges of $K$. This sharpness can be empirically observed as well in Figure 1; the details are stated in Section 6.2.

## 5    RELATED WORK

While our work and the existing works reviewed in Section 4 attempt to understand CURL by connecting the contrastive loss to the mean supervised loss, there are other approaches; Wang & Isola (2020) showed that the contrastive loss asymptotically favors data representation uniformly distributed over the unit sphere yet aligning across semantically similar samples. Li et al. (2021) proposed an alternative loss function to the contrastive loss based on a kernel metric, following the

---

[6]More precisely, Arora et al. (2019) bound the averaged supervised loss over a part of the latent classes rather than $R_{\mu\text{-supv}}$. Thus we can obtain a slightly better upper bound than Arora et al.'s bound shown in Table 1. Nevertheless, the scale of the upper bound is dominated by the coefficient $(1 - \tau_K)v_{K+1}$.

[7]Note that Nozawa & Sato (2021)'s bound also implies larger $K$ is better. Still, its mechanism and resulting explainability are different from ours. See Appendix D for the further discussions.

similar idea to Wang & Isola (2020). Tosh et al. (2021) showed that a (linear) mean classifier learned in CURL can approximate the (potentially nonlinear) Bayes classifier well.

While our work does not handle data augmentation (DA), several works analyzed how DA affects the performance of downstream classification. Wen & Li (2021) showed that DA is necessary to recover sparse signals under a specific assumption on the model architecture. HaoChen et al. (2021) introduces a notion of the augmentation graph, representing how likely the nearby samples are generated via DA and showed that a type of contrastive loss could be viewed as a low-rank approximation of the adjacency matrix of the augmentation graph. von Kügelgen et al. (2021) proposed a loss function that enables the model to identify invariant factors across DA.

We mention a few works analyzing the other types of self-supervised learning; Garg & Liang (2020) analyzed masked self-supervised learning, Wei et al. (2021) analyzed the input consistency loss for unsupervised learning, and Saunshi et al. (2021) analyzed auto-regressive language models.

As a final remark, multi-sample estimators (Oord et al., 2018; Poole et al., 2019; Song & Ermon, 2020) popularly used in *mutual information (MI) estimation* are substantially related to the contrastive loss. Although the multi-sample estimators have high bias and low variance compared to variational estimators in general (Poole et al., 2019; Song & Ermon, 2020; Guo et al., 2021), the quantitative analysis of the bias-variance trade-off of the multi-sample MI estimators has yet to be clearly known.[8] Note that Tian et al. (2020) and Tschannen et al. (2020) experimentally showed that maximizing *tighter* MI bound does not necessarily lead to good representation; there is no guarantee that the model can achieve higher MI by the tighter bound.

## 6 EXPERIMENTS

We verified our theoretical findings with experiments on synthetic datasets (Section 6.1), vision, and language datasets (Section 6.2). The details of the experimental setup are provided in Appendix F.

### 6.1 SMALL-SCALE EXPERIMENTS ON SYNTHETIC DATASET

**Dataset and learning setups.**    We create a synthetic dataset `circle`, which is a 2D dataset created as follows: for each class $c \in [C]$, $1\,000$ samples are drawn from $\text{Uniform}([-0.5 \; -0.5], [0.5, 0.5])$, normalized, and multiplied by $c+1/2$. The generated samples are nonlinear and require disentanglement to be linearly separable. We treated $60\%$ of the generated samples as a training dataset and the rest of the samples as a test dataset. We did not use data augmentation.

As a feature extractor $\mathbf{f}$, we used a multi-layer perceptron (the number of units 2-256-256-256) with the ReLU activation functions following after each hidden layer. During the training, the extracted feature representations are normalized. For negative samples, we sampled $K \in \{1, 4, 16, 64, 256\}$ samples without replacement from $2B - 2$ points included in the same mini-batch to avoid the influence of mini-batch size $B$, inspired by Ash et al. (2021).[9]

**Results.**    Figure 5 shows a single trajectory in the $(R_{\text{cont}}, R_{\mu\text{-supv}})$-plot and the feasible region (confer Figure 2) for each $K$. We plotted the trajectories by tracking $(R_{\text{cont}}(\mathbf{f}^{(t)}), R_{\mu\text{-supv}}(\mathbf{f}^{(t)}))$ at each epoch $t$ computed with the test dataset. All trajectories were located in between the upper $(R_{\text{cont}} + \Delta_{\text{U}})$ and lower $(R_{\text{cont}} + \Delta_{\text{L}})$ bounds as a matter of course. Given that the existing learning bounds provide the much larger upper bounds (Figure 4), our learning bounds provide the finest estimate of the mean supervised loss. In addition, it is remarkable that all trajectories have nearly the same slopes as our learning bounds, which constitutes solid evidence that our learning bounds capture the learning dynamics well.

In Figure 6, the mean supervised loss and accuracy are compared with the different $K$. For each $K$, we conducted the same experiments with eight different random seeds, and the standard deviations are shown in the figures. From these figures, it can be concluded that the contrastive loss performance

---

[8]There are some quantitative analyses such as the bias of general MI estimators (Gao et al., 2015; McAllester & Stratos, 2020) and the variance of the NWJ/MINE estimators (Song & Ermon, 2020). We will further discuss the implication of the bias analysis in Appendix E.

[9]Each mini-batch consists of $B$ pairs of positive pairs. The candidates of the negative samples are the $2B - 2$ samples excluding the anchor and its paired point.

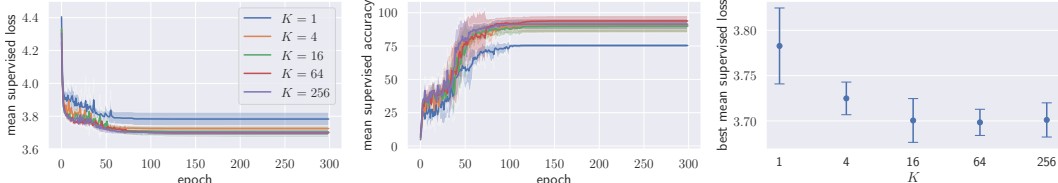

Figure 5: Learning trajectories of the `circle` dataset in the $(R_{\text{cont}}, R_{\mu\text{-supv}})$-plot. The trajectories are plotted with gradient color lines, indicating the epochs.

Figure 6: For each $K$, eight runs on the `circle` dataset are averaged with the standard deviations plotted. **(Left)** the test mean supervised losses at each epoch with the different negative sample sizes $K$. **(Middle)** the test mean supervised accuracy at each epoch with the different negative sample sizes $K$. **(Right)** the best test mean supervised loss with the different negative sample sizes $K$.

becomes better with the larger $K$ in the sense that the supervised loss improved and the variance shrank. The variance improvement can be theoretically observed from Figure 5 as well; the larger $K$ is, the smaller the gap between upper and lower bounds becomes.

## 6.2 LARGE-SCALE EXPERIMENTS ON VISION AND LANGUAGE DATASETS

We used the same datasets as Arora et al. (2019): CIFAR-100 (Krizhevsky, 2009) and Wiki-3029 (Arora et al., 2019) datasets. In addition, we used CIFAR-10 (Krizhevsky, 2009) dataset.

**Learning setups.** To create positive pairs for contrastive learning, we treated the supervised classes as latent classes as in Arora et al. (2019) and Ash et al. (2021). We treated the original supervised classes of CIFAR-10/100 as $[C]$; $C = 10$ and $C = 100$, respectively. We used $K$ in $\{4, 16, 32, 64, 128, 512\}$ and in $\{4, 64, 128, 512\}$ for CIFAR-10/100, respectively. For Wiki-3029, we used $C \in \{500, 1\,000, 2\,000, 3\,029\}$ and $K \in \{8, 64, 256, 1\,024\}$. For each different $(C, K)$ pair, we trained the feature extractor $\mathbf{f}$ on the training dataset. We then evaluated its supervised performance on the test dataset with mean and linear classifiers. We used ResNet-18 (He et al., 2016)-based feature extractor $\mathbf{f}$ for CIFAR-10/100 and the fasttext (Joulin et al., 2017)-based feature extractor for Wiki-3029.

**Results.** Figure 1 shows the comparison between the estimated upper bounds using Theorem 1 and actual supervised loss on CIFAR-10 for different $K$ on the test dataset. We estimated the bounds by substituting the actual $R_{\text{cont}}(\mathbf{f})$ to the equations shown in Theorem 1 and Table 1. Our learning bound gave the closest upper bound to the experimental value of the supervised loss. The existing learning bounds of Arora et al. (2019) and Ash et al. (2021) were not sharp enough to explain the classification performance. Although Nozawa & Sato's bound was comparable with our bound, it was valid only in $K + 1 \geq C$ and tended to increase near $K + 1 = C$, as we explained in Section 4.

We investigated how $K$ affects the test accuracy for different $C$ in Figure 7. The test accuracy improved or was saturated with the larger $K$ for all $C$ on Wiki-3029. In contrast, it was degraded as $K$ increased in mean and linear classifiers on CIFAR-10/100. This behavior could be partly because of the gap between the cross-entropy loss and the supervised accuracy—the theory of CURL, including the existing studies, usually focuses on the cross-entropy loss only. Figure 9 in Appendix F.6 revealed that the supervised loss was not significantly worse with the larger $K$ on CIFAR-10/100.

With the smaller $K$ and large $C$, we found that long epochs were more effective to improve classification accuracy than increasing the negative sample size $K$ (Figure 8b). While similar experimental results were reported by Chen et al. (2020, Figure 9), it is important to remark that we randomly

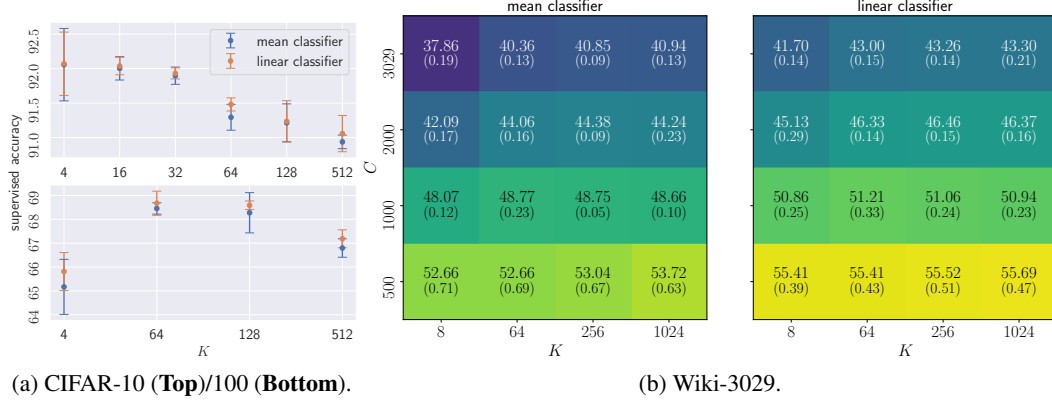

(a) CIFAR-10 (**Top**)/100 (**Bottom**).

(b) Wiki-3029.

Figure 7: Mean and linear classifier's test accuracy on CIFAR-10/100 and Wiki-3029 when varying the negative samples size $K$. For Wiki-3029, we also change the number of latent classes $C$. The error bars in (**a**) and parenthesized number in (**b**) indicate the standard deviation of three runs.

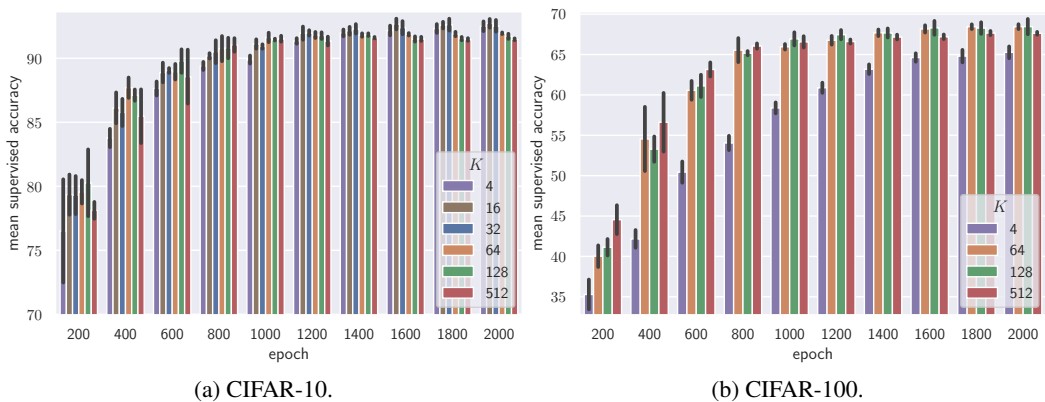

(a) CIFAR-10.

(b) CIFAR-100.

Figure 8: Test accuracy of mean classifier at every 200 epochs on CIFAR-10/100. In CIFAR-100, the accuracy of $K = 4$ and the others' have a large gap at smaller epochs at epoch 200, but the gap become smaller when epochs increase. The error bars indicate the standard deviation of three runs.

drew $K$ negative samples from the $2B - 2$ samples in the given mini-batch at each iteration as in Ash et al. (2021)—a different approach was used by Chen et al. (2020) to regard the all samples in the mini-batch except an anchor sample as negative samples. Under our experimental setup, a learner may encounter less diverse samples with the smaller $K$ even if the mini-batch size $B$ is the same, which could make the performance of downstream classification worse—the longer epochs are necessary to mitigate the issue. Since the CIFAR-10 dataset has a smaller $C$ and is simpler than the CIFAR-100, all accuracies were saturated with similar epochs for all $K$ (Figure 8a).

## 7 CONCLUSION

In this study, we established novel theoretical bounds for contrastive unsupervised representation learning. We derived sharp upper and lower bounds on a downstream classification loss that are tight in the negative sample size. They suggest that the contrastive loss can be used as a downstream loss estimator and its estimation gap decays with larger negative samples. In the experiments, we verified that our bounds well explained learning dynamics on the synthetic dataset, and the estimation gap was large with smaller negative samples. In addition, the empirical downstream classification loss was best explained by our learning bound compared to the existing ones. It is left open to fill the gap between the supervised loss and accuracy. In addition, we may understand contrastive unsupervised representation learning better by incorporating the optimization perspective into the theoretical analysis.

ETHICS STATEMENT

Since this study is theoretical in nature, we do not have specific ethical concerns.

REPRODUCIBILITY STATEMENT

All experimental codes are available in the supplementary material. In addition, the details of experimental datasets and hyper-parameters of learning algorithms are described in Appendix F.

All theorems' proofs are described in Appendix B. Appendix A provides Lemmas used in Appendix B. Appendices C and D provide how to derive equations used in Section 3.2 and Section 4, respectively.

«APPENDIX»

# SHARP LEARNING BOUNDS FOR CONTRASTIVE UNSUPERVISED REPRESENTATION LEARNING

As an additional notation, the $d$-dimensional ball of radius $r$ associated with the $\mathbb{L}_p$-norm is denoted by $\mathbb{B}_p^d(r) := \{\mathbf{x} \in \mathbb{R}^d \mid \|\mathbf{x}\|_p \leq r\}$. For $\mathbf{z} \in \mathbb{R}^C$, the log-sum-exp function is denoted by $\text{LSE}(\mathbf{z}) := \ln(\sum_{c \in [C]} \exp(z_c))$.

## A USEFUL LEMMAS

In this section, a few lemmas are introduced in order to prove the main results.

**Lemma 3.** *For $\mathbf{z} \in \mathbb{R}^N$,*

$$\ln\left(1 + \sum_{n \in [N]} \exp(-z_i)\right) \geq -\ln\left(1 + \sum_{n \in [N]} \exp(z_k)\right) + \Delta_0, \tag{10}$$

*where $\Delta_0 := 2\ln(1 + N)$.*

*Proof.* Define $H(\mathbf{z}) := \text{LSE}(-[0\ z_1\ \ldots\ z_N]) + \text{LSE}([0\ z_1\ \ldots\ z_N])$. Then, $\Delta_0$ is a lower bound of $H$. Since

$$\frac{\partial H}{\partial z_i} = -\frac{\exp(-z_i)}{\sum_{n \in [N]} \exp(-z_n)} + \frac{\exp(z_i)}{\sum_{n \in [N]} \exp(z_n)} \tag{11}$$

for all $i \in [N]$, $\mathbf{z} = \mathbf{0}_N$ satisfies the first-order optimality condition of $H$. By noting that $H$ is convex due to the convexity of the log-sum-exp functions, $H$ is minimized at $\mathbf{z} = \mathbf{0}_N$. Hence, we can choose $\Delta_0 = H(\mathbf{0}_N) = 2\ln(1 + N)$. $\qquad\square$

**Lemma 4.** *For $z_0 \in [-L^2, L^2]$, $\mathbf{z} \in [-L^2, L^2]^N$, and $\rho > 0$,*

$$\ln\frac{\exp(-z_0)}{(1+\rho)\exp(-z_0) + \rho\sum_{n \in [N]}\exp(-z_n)} \geq -\ln\frac{\exp(z_0)}{(1+\rho)\exp(z_0) + \rho\sum_{n \in [N]}\exp(z_n)} - \Delta_1, \tag{12}$$

*where $\Delta_1 := 2\ln\left(\rho\left(N + 1 + \frac{1}{\rho}\right)\cosh(L^2)\right)$.*

*Proof.* The goal is to find a tight upper bound $\Delta_1$ of

$$-\ln\frac{\exp(z_0)}{(1+\rho)\exp(z_0) + \rho\sum_{n \in [N]}\exp(z_n)} - \ln\frac{\exp(-z_0)}{(1+\rho)\exp(-z_0) + \rho\sum_{n \in [N]}\exp(-z_n)} \tag{13}$$

$$= \ln\left\{(1+\rho)\exp(z_0) + \rho\sum_{n \in [N]}\exp(z_n)\right\} + \ln\left\{(1+\rho)\exp(-z_0) + \rho\sum_{n \in [N]}\exp(-z_n)\right\} \tag{14}$$

$$= 2\ln\rho + \underbrace{\ln\left\{\left(1+\frac{1}{\rho}\right)\exp(z_0) + \sum_n\exp(z_n)\right\} + \ln\left\{\left(1+\frac{1}{\rho}\right)\exp(-z_0) + \sum_n\exp(-z_n)\right\}}_{:=H'(\tilde{\mathbf{z}})}, \tag{15}$$

where $\tilde{\mathbf{z}} := [z_0\ \mathbf{z}^\top]^\top$.

Equivalently, we aim at an upper bound of $H'(\tilde{\mathbf{z}})$ for $\tilde{\mathbf{z}} \in \mathbb{B}_\infty^{N+1}(L^2)$. Observe that $H'(\tilde{\mathbf{z}}) = \text{LSE}\left(\tilde{\mathbf{z}} + \left[\ln\left(1 + \frac{1}{\rho}\right)\ 0\ \ldots\ 0\right]^\top\right) + \text{LSE}\left(-\tilde{\mathbf{z}} + \left[\ln\left(1 + \frac{1}{\rho}\right)\ 0\ \ldots\ 0\right]^\top\right)$ is the sum of the two log-sum-exp functions after adding the constant vector hence it is convex in $\tilde{\mathbf{z}}$. In addition, the domain $\mathbb{B}_\infty^{N+1}(L^2)$ is a compact convex polytope. Henceforth, every vertex of the polytope, $\tilde{\mathbf{z}} \in \{-L^2, L^2\}^{N+1}$, is a local maximizer because maximizing $H'(\tilde{\mathbf{z}})$ is concave minimization over a convex polytope (Pardalos & Rosen, 1986). Since $H'(\tilde{\mathbf{z}})$ is symmetric in every $z_n$ except $z_0$, we need to divide the cases depending on either $z_0 = L^2$ or $z_0 = -L^2$ and compare the maximum values of $H'(\tilde{\mathbf{z}})$.

Under line $z_0 = L^2$: In order to maximize $H'(\tilde{\mathbf{z}})$, it is sufficient to test the vertices $\tilde{\mathbf{z}} \in \{\mathbf{z} \in \mathbb{B}_\infty^{N+1}(L^2) \mid z_0 = L^2\}$ and see the difference between $\tilde{H}'(m)$ and $\tilde{H}'(m+1)$, where

$$\tilde{H}'(m) := H'\Big(L^2, \underbrace{L^2, \ldots, L^2}_{m}, \underbrace{-L^2, \ldots, -L^2}_{N-m}\Big) \tag{16}$$

$$= \ln\left\{\left(m + 1 + \frac{1}{\rho}\right)\exp(-L^2) + (N - m)\exp(L^2)\right\}$$

$$+ \ln\left\{\left(m + 1 + \frac{1}{\rho}\right)\exp(L^2) + (N - m)\exp(-L^2)\right\} \tag{17}$$

for $m \in \{0, \ldots, N\}$ to seek the maximum. Here, the definition of $\tilde{H}'$ can be naturally extended over the real domain $m \in \mathbb{R}$. A simple algebra shows

$$\exp\left(\tilde{H}'(m)\right) = \underbrace{\left(2 - (\exp(2L^2) + \exp(-2L^2))\right)}_{<0 \text{ (AM-GM inequality)}}\left(m - \frac{N - 1 - \frac{1}{\rho}}{2}\right)^2 + \text{Const.} \tag{18}$$

Since $\tilde{H}'$ is concave over $m \in \mathbb{R}$, $\tilde{H}'$ is maximized at $m = \left[\frac{N-1-\frac{1}{\rho}}{2}\right]_+ (< N)$. If $\rho < \frac{1}{N-1}$, then the maximum value is

$$\tilde{H}'\left(\frac{N - 1 - \frac{1}{\rho}}{2}\right) = 2\ln\left(\left(N + 1 + \frac{1}{\rho}\right)\cosh(L^2)\right). \tag{19}$$

If $\rho \geq \frac{1}{N-1}$, $\tilde{H}'$ is maximized at $m = 0$ and its value is

$$\tilde{H}'(0) = \ln\left(N^2 + \left(1 + \frac{1}{\rho}\right)^2 + N\left(1 + \frac{1}{\rho}\right)(\exp(2L^2) + \exp(-2L^2))\right) \tag{20}$$

$$\leq \ln\left(\frac{\exp(2L^2) + \exp(-2L^2)}{2}\left(N^2 + \left(1 + \frac{1}{\rho}\right)^2 + 2N\left(1 + \frac{1}{\rho}\right)\right)\right) \tag{21}$$

$$= \ln\left(\left(N + 1 + \frac{1}{\rho}\right)\cosh(2L^2)\right), \tag{22}$$

where the inequality follows from the AM-GM inequality $\frac{\exp(2L^2) + \exp(-2L^2)}{2} \geq 1$.

Under line $z_0 = -L^2$: Similarly, we define

$$\tilde{H}'(m) := H'\Big(-L^2, \underbrace{L^2, \ldots, L^2}_{m}, \underbrace{-L^2, \ldots, -L^2}_{N-m}\Big) \tag{23}$$

$$= \ln\left\{m\exp(-L^2) + \left(N + 1 + \frac{1}{\rho} - m\right)\exp(L^2)\right\}$$

$$+ \ln\left\{m\exp(L^2) + \left(N + 1 + \frac{1}{\rho} - m\right)\exp(-L^2)\right\}, \tag{24}$$

and extends its definition over the real line $m \in [0, N]$. Since

$$\exp\left(\tilde{H}'(m)\right) = \underbrace{(2 - (\exp(2L^2) + \exp(-2L^2)))}_{<0 \text{ (AM-GM inequality)}} \left(m - \frac{N+1+\frac{1}{\rho}}{2}\right)^2 + \text{Const}, \qquad (25)$$

$\tilde{H}'$ is maximized at $m = \min\left\{\frac{N+1+\frac{1}{\rho}}{2}, N\right\} (> 0)$. If $\rho \geq \frac{1}{N-1}$, $\tilde{H}'$ is maximized at $m = \frac{N+1+\frac{1}{\rho}}{2}$ and its value is

$$\tilde{H}'\left(\frac{N+1+\frac{1}{\rho}}{2}\right) = 2\ln\left(\left(N+1+\frac{1}{\rho}\right)\cosh(L^2)\right). \qquad (26)$$

If $\rho < \frac{1}{N-1}$, $\tilde{H}'$ is maximized at $m = N$ and its value is

$$\tilde{H}'(N) = \ln\left(N^2 + \left(1+\frac{1}{\rho}\right)^2 + N\left(1+\frac{1}{\rho}\right)(\exp(2L^2) + \exp(-2L^2))\right) \qquad (27)$$

$$\leq \ln\left(\left(N+1+\frac{1}{\rho}\right)\cosh(2L^2)\right). \qquad (28)$$

After all, $\tilde{H}'$ is bounded from above by either

$$2\ln\left(\left(N+1+\frac{1}{\rho}\right)\cosh(L^2)\right) \text{ or } \ln\left(\left(N+1+\frac{1}{\rho}\right)\cosh(2L^2)\right). \qquad (29)$$

These two values can be compared as follows.

$$2\ln\left(\left(N+1+\frac{1}{\rho}\right)\cosh(L^2)\right) - \ln\left(\left(N+1+\frac{1}{\rho}\right)\cosh(2L^2)\right) \qquad (30)$$

$$= \ln\left(N+1+\frac{1}{\rho}\right) + \ln\cosh^2(L^2) - \ln\cosh(2L^2) \qquad (31)$$

$$= \ln\left(N+1+\frac{1}{\rho}\right) + \ln\frac{1+\frac{1}{\cosh(2L^2)}}{2} \qquad (32)$$

$$\geq \ln\left(N+1+\frac{1}{\rho}\right) \qquad (33)$$

$$> 0, \qquad (34)$$

where the first inequality is due to the AM-GM inequality $\frac{\exp(2L^2)+\exp(-2L^2)}{2} \geq 1$. Hence, regardless of $\rho$, $2\ln\left(\left(N+1+\frac{1}{\rho}\right)\cosh(L^2)\right)$ is a tight upper bound of $\tilde{H}'$. $\qquad\square$

**Lemma 5.** *For all $z_0 \in \mathbb{R}$ and $\mathbf{z} \in \mathbb{R}^K$ such that $z_0, z_k \in [-L^2, L^2]$ ($\forall k \in [K]$),*

$$\ln\frac{\exp(-z_0)}{\exp(-z_0) + \sum_{k\in[K]}\exp(-z_k)}$$

$$\geq -\ln\frac{\exp(z_0)}{\exp(z_0) + \sum_{k\in[K]}\exp(z_k)} - 2\ln\left\{(K+1)\cosh(L^2)\right\}. \qquad (35)$$

*Proof.* We write $\tilde{\mathbf{z}} := [z_0 \ \mathbf{z}^\top]^\top$. Let $H(\tilde{\mathbf{z}})$ be a function such that

$$H(\tilde{\mathbf{z}}) := -\ln\frac{\exp(z_0)}{\exp(z_0) + \sum_{k\in[K]}\exp(z_k)} - \ln\frac{\exp(-z_0)}{\exp(-z_0) + \sum_{k\in[K]}\exp(-z_k)} \qquad (36)$$

$$= \ln\sum_{k=1}^{K+1}\exp(\tilde{z}_k) + \ln\sum_{k=1}^{K+1}\exp(-\tilde{z}_k). \qquad (37)$$

Our goal is to find a tight upper bound of $H(\tilde{\mathbf{z}})$ for $\tilde{\mathbf{z}} \in \mathbb{B}_{\infty}^{K+1}(L^2)$.

Observe that $H(\tilde{\mathbf{z}})$ is the sum of the two log-sum-exp functions hence it is convex in $\tilde{\mathbf{z}}$. In addition, the domain $\mathbb{B}_{\infty}^{K+1}(L^2)$ is a compact convex polytope. Henceforth, every vertex of the polytope, $\tilde{\mathbf{z}} \in \{-L^2, L^2\}^{K+1}$, is a local maximizer because maximizing $H(\tilde{\mathbf{z}})$ is concave minimization over a convex polytope (Pardalos & Rosen, 1986). Since $H(\tilde{\mathbf{z}})$ is symmetric in every element $\tilde{z}_k$, it is sufficient to test the vertices and see the difference between

$$H\left( \underbrace{(L^2, \ldots, L^2}_{\# = j}, -L^2, \ldots, -L^2)\right) \text{ and } H\left( \underbrace{(L^2, \ldots, L^2}_{\# = j+1}, -L^2, \ldots, -L^2)\right) \tag{38}$$
$$\underbrace{\phantom{H\left((L^2, \ldots, L^2, -L^2, \ldots, -L^2)\right)}}_{:=\tilde{H}(j)} \qquad \underbrace{\phantom{H\left((L^2, \ldots, L^2, -L^2, \ldots, -L^2)\right)}}_{:=\tilde{H}(j+1)}$$

for $j \in \{0, \ldots, K\}$ to seek out the global maximum. For $0 \leq j \leq K$, a simple algebra shows

$$\exp\left(\tilde{H}(j)\right) - \exp\left(\tilde{H}(j+1)\right) = (K - 2j) \underbrace{\left\{2 - \left(\exp(2L^2) + \exp(-2L^2)\right)\right\}}_{\substack{\leq 0 \\ \text{because of AM-GM inequality}}}, \tag{39}$$

from which we can tell that $\tilde{H}(j)$ is maximized at $j = K/2$ when $K$ is even and $j = (K+1)/2$ when $K$ is odd. In addition, it is confirmed that

$$\exp\left(\tilde{H}\left(\frac{K}{2}\right)\right) - \exp\left(\tilde{H}\left(\frac{K+1}{2}\right)\right) = \frac{2 - \left(\exp(2L^2) + \exp(-2L^2)\right)}{4} \tag{40}$$
$$\leq 0, \tag{41}$$

where the AM-GM inequality is invoked at the last line. Eventually, $\tilde{H}((K+1)/2)$ turns out to be a tight upper bound of $H(\tilde{\mathbf{z}})$ for $\tilde{\mathbf{z}} \in \mathbb{B}_{\infty}^{K+1}(L^2)$. It is elementary to confirm $\tilde{H}((K+1)/2) = 2 \ln\left\{(K+1)\cosh(L^2)\right\}$. $\qquad \square$

## B    PROOFS OF MAIN RESULTS

In this section, we provide proofs for the main results, Theorems 1 and 2.

**Theorem 1.** *For all $\mathbf{f}$ such that $\|\mathbf{f}(\mathbf{x})\|_2 \leq L$ ($\forall \mathbf{x} \in \mathcal{X}$), the following inequality holds.*

$$R_{\mu\text{-supv}}(\mathbf{f}) \leq R_{\text{cont}}(\mathbf{f}) + \Delta_{\text{U}}, \tag{3}$$

*where $\Delta_{\text{U}} \coloneqq 2\ln((CK\pi_{(-1)} + 1)\cosh(L^2)) - 2\ln(1 + K) - \ln K\pi_{(-1)}$.*

*Proof of Theorem 1.* Define the two constants

$$\Delta_0 \coloneqq 2\ln(1 + K), \tag{42}$$
$$\Delta_1 \coloneqq 2\ln((CK\pi_{(-1)} + 1)\cosh(L^2)). \tag{43}$$

The proof begins with the Jensen's inequality applied on the contrastive loss:

$$R_{\text{cont}}(\mathbf{f}) = \mathop{\mathbb{E}}_{c^+, \{c_k^-\}, \mathbf{x}, \mathbf{x}^+, \{\mathbf{x}_k^-\}} \ln\left(1 + \sum_{k \in [K]} \exp(\mathbf{f}(\mathbf{x})^\top (\mathbf{f}(\mathbf{x}_k^-) - \mathbf{f}(\mathbf{x}^+)))\right) \tag{44}$$

$$\geq \mathop{\mathbb{E}}_{c^+, \{c_k^-\}, \mathbf{x}} \ln\left(1 + \sum_{k \in [K]} \exp(\mathbf{f}(\mathbf{x})^\top (\boldsymbol{\mu}_{c_k^-} - \boldsymbol{\mu}_{c^+}))\right) \tag{45}$$

$$\geq - \mathop{\mathbb{E}}_{c^+, \{c_k^-\}, \mathbf{x}} \ln\left(1 + \sum_{k \in [K]} \exp(\mathbf{f}(\mathbf{x})^\top (\boldsymbol{\mu}_{c^+} - \boldsymbol{\mu}_{c_k^-}))\right) + \Delta_0, \tag{46}$$

where the second inequality follows from Lemma 3. The first term, the origin-symmetric transform of the contrastive loss, is further bounded as follows.

$$
- \mathop{\mathbb{E}}_{c^+, \{c_k^-\}, \mathbf{x}} \ln \left( 1 + \sum_{k \in [K]} \exp(\mathbf{f}(\mathbf{x})^\top (\boldsymbol{\mu}_{c^+} - \boldsymbol{\mu}_{c_k^-})) \right)
$$

$$
\geq - \mathop{\mathbb{E}}_{c^+, \mathbf{x}} \ln \left( 1 + \sum_{k \in [K]} \mathop{\mathbb{E}}_c \exp(\mathbf{f}(\mathbf{x})^\top (\boldsymbol{\mu}_{c^+} - \boldsymbol{\mu}_c)) \right) \tag{47}
$$

$$
\geq - \mathop{\mathbb{E}}_{c^+, \mathbf{x}} \ln \left( 1 + K\pi_{(-1)} \left( 1 + \sum_{c \in [C] \setminus \{c^+\}} \exp(\mathbf{f}(\mathbf{x})^\top (\boldsymbol{\mu}_{c^+} - \boldsymbol{\mu}_c)) \right) \right), \tag{48}
$$

where the first inequality is the Jensen's inequality and the second inequality holds by noting that $\mathbb{E}_c[A] = \sum_c \pi_c A \geq \sum_c \pi_{(-1)} A$. In order to proceed further, the origin-symmetric transform is needed to be applied on the lower-bounding term, resulting in a term aligning to the mean supervised loss.

$$
- \mathop{\mathbb{E}}_{c^+, \mathbf{x}} \ln \left( 1 + K\pi_{(-1)} \left( 1 + \sum_{c \in [C] \setminus \{c^+\}} \exp(\mathbf{f}(\mathbf{x})^\top (\boldsymbol{\mu}_{c^+} - \boldsymbol{\mu}_c)) \right) \right)
$$

$$
\geq \mathop{\mathbb{E}}_{c^+, \mathbf{x}} \ln \left( 1 + K\pi_{(-1)} \left( 1 + \sum_{c \in [C] \setminus \{c^+\}} \exp(\mathbf{f}(\mathbf{x})^\top (\boldsymbol{\mu}_c - \boldsymbol{\mu}_{c^+})) \right) \right) - \Delta_1 \tag{49}
$$

$$
= \mathop{\mathbb{E}}_{c^+, \mathbf{x}} \ln \left( \frac{1}{K\pi_{(-1)}} + 1 + \sum_{c \in [C] \setminus \{c^+\}} \exp(\mathbf{f}(\mathbf{x})^\top (\boldsymbol{\mu}_c - \boldsymbol{\mu}_{c^+})) \right) - \Delta_1 + \ln K\pi_{(-1)} \tag{50}
$$

$$
\geq \mathop{\mathbb{E}}_{c^+, \mathbf{x}} \ln \left( 1 + \sum_{c \in [C] \setminus \{c^+\}} \exp(\mathbf{f}(\mathbf{x})^\top (\boldsymbol{\mu}_c - \boldsymbol{\mu}_{c^+})) \right) - \Delta_1 + \ln K\pi_{(-1)} \tag{51}
$$

$$
= R_{\mu\text{-supv}}(\mathbf{f}) - \Delta_1 + \ln K\pi_{(-1)}, \tag{52}
$$

where the first inequality is a consequence of Lemma 4 and the second inequality is simply observed by $\ln(\frac{1}{K\pi_{(-1)}} + x) \geq \ln x$ for any $x > 0$ since $\frac{1}{K\pi_{(-1)}} \geq 0$. Combining all the above, we get the final bound $R_{\mu\text{-supv}}(\mathbf{f}) \leq R_{\text{cont}}(\mathbf{f}) - \Delta_0 + \Delta_1 - \ln K\pi_{(-1)}$. $\qquad\square$

Even if Theorem 1 assumes $\mathcal{Y} = [C]$, it can be extended to the cases $\mathcal{Y} \subseteq [C]$ (subset) and $\mathcal{Y}$ is a coarse-grained set of $[C]$. When $\mathcal{Y} \neq [C]$, $R_{\mu\text{-supv}}$ is defined over the class set $\mathcal{Y}$, while $R_{\text{cont}}$ is defined over the class set $[C]$. In the subset case, we can replace Eq. (52) with the subset class bound:

$$
\mathop{\mathbb{E}}_{c^+, \mathbf{x}} \ln \left( 1 + \sum_{c \in [C] \setminus \{c^+\}} \exp(\mathbf{f}(\mathbf{x})^\top (\boldsymbol{\mu}_c - \boldsymbol{\mu}_{c^+})) \right)
$$

$$
\geq \mathop{\mathbb{E}}_{c^+, \mathbf{x}} \ln \left( 1 + \sum_{c \in \mathcal{Y} \setminus \{c^+\}} \exp(\mathbf{f}(\mathbf{x})^\top (\boldsymbol{\mu}_c - \boldsymbol{\mu}_{c^+})) \right) = R_{\mu\text{-supv}}(\mathbf{f}). \tag{53}
$$

In the coarse-grained case, Eq. (52) holds without modification.

**Theorem 2.** *For all $\mathbf{f}$ such that $\|\mathbf{f}(\mathbf{x})\|_2 \leq L \ (\forall \mathbf{x} \in \mathcal{X})$, the following inequality holds.*

$$
R_{\mu\text{-supv}}(\mathbf{f}) \geq R_{\text{cont}}(\mathbf{f}) + \Delta_{\mathrm{L}}, \tag{4}
$$

*where $\Delta_{\mathrm{L}} := \mathbb{H}(\boldsymbol{\pi}) + \ln \frac{K}{(K+1)^2} - 2\ln\cosh(L^2)$.*

*Proof of Theorem 2.* The proof is essentially a consequence of the Fenchel's inequality and the Jensen's inequality. First, by noting that the convex conjugate of the log-sum-exp function is the

negative Shannon entropy, the following identity is obtained.

$$R_{\mu\text{-supv}}(\mathbf{f}) = \underset{\mathbf{x},y}{\mathbb{E}} \left[ -\mathbf{f}(\mathbf{x})^\top \boldsymbol{\mu}_y + \mathrm{LSE}\left(\mathbf{W}^\mu \mathbf{f}(\mathbf{x})\right) \right] \tag{54}$$

$$= \underset{\mathbf{x},y}{\mathbb{E}} \left[ -\mathbf{f}(\mathbf{x})^\top \boldsymbol{\mu}_y + \sup_{\mathbf{p} \in \triangle^C} \left\{ \mathbf{p}^\top (\mathbf{W}^\mu \mathbf{f}(\mathbf{x})) + \mathbb{H}\left(\mathbf{p}\right) \right\} \right]. \tag{55}$$

If we choose an arbitrary $\mathbf{p} \in \triangle^C$, $R_{\mu\text{-supv}}(\mathbf{f})$ is lower bounded (Fenchel's inequality). Our choice is $\mathbf{p} = \boldsymbol{\pi}$. Recall that $K$ is the number of negative samples. Then,

$$R_{\mu\text{-supv}}(\mathbf{f}) \geq \underset{c^+,\mathbf{x}}{\mathbb{E}} \left[ -\mathbf{f}(\mathbf{x})^\top \boldsymbol{\mu}_{c^+} + \sum_{c^- \in \mathcal{Y}} \pi_{c^-} \mathbf{f}(\mathbf{x})^\top \boldsymbol{\mu}_{c^-} \right] + \mathbb{H}\left(\boldsymbol{\pi}\right) \tag{56}$$

$$= \underset{c^+}{\mathbb{E}} \underset{\mathbf{x},\mathbf{x}^+ \sim \mathcal{D}^2_{c^+}}{\mathbb{E}} \left[ -\mathbf{f}(\mathbf{x})^\top \mathbf{f}(\mathbf{x}^+) + \mathbf{f}(\mathbf{x})^\top \left( \underset{c^-}{\mathbb{E}} \underset{\mathbf{x}^- \sim \mathcal{D}_{c^-}}{\mathbb{E}} \left[ \mathbf{f}(\mathbf{x}^-) \right] \right) \right] + \mathbb{H}\left(\boldsymbol{\pi}\right) \tag{57}$$

$$= \underset{c^+}{\mathbb{E}} \underset{\mathbf{x},\mathbf{x}^+}{\mathbb{E}} \left[ -\mathbf{f}(\mathbf{x})^\top \mathbf{f}(\mathbf{x}^+) + \frac{1}{K} \sum_{k \in [K]} \underset{c_k^-}{\mathbb{E}} \underset{\mathbf{x}_k^-}{\mathbb{E}} [\mathbf{f}(\mathbf{x})^\top \mathbf{f}(\mathbf{x}_k^-)] \right] + \mathbb{H}\left(\boldsymbol{\pi}\right) \tag{58}$$

$$= \underset{c^+,\{c_k^-\}_k}{\mathbb{E}} \underset{\mathbf{x},\mathbf{x}^+,\{\mathbf{x}_k^-\}_k}{\mathbb{E}} \left[ -\frac{1}{K} \sum_{k \in [K]} \left( \mathbf{f}(\mathbf{x})^\top \mathbf{f}(\mathbf{x}^+) - \mathbf{f}(\mathbf{x})^\top \mathbf{f}(\mathbf{x}_k^-) \right) \right] + \mathbb{H}\left(\boldsymbol{\pi}\right) \tag{59}$$

$$= \underset{c^+,\{c_k^-\}_k}{\mathbb{E}} \underset{\mathbf{x},\mathbf{x}^+,\{\mathbf{x}_k^-\}_k}{\mathbb{E}} \left[ -\frac{1}{K} \sum_{k \in [K]} \ln \exp(\mathbf{f}(\mathbf{x})^\top (\mathbf{f}(\mathbf{x}^+) - \mathbf{f}(\mathbf{x}_k^-))) \right] + \mathbb{H}\left(\boldsymbol{\pi}\right). \tag{60}$$

Here, we can proceed with the Jensen's inequality to lower bound the first term: for a non-negative vector $\mathbf{z} \in \mathbb{R}^N_{\geq 0}$, the inequality $-N^{-1} \sum_{i \in [N]} \ln z_i \geq -\ln(N^{-1} \sum_{i \in [N]} z_i)$ holds. If we set $z_k = \exp(\mathbf{f}(\mathbf{x})^\top (\mathbf{f}(\mathbf{x}^+) - \mathbf{f}(\mathbf{x}_k^-)))$ for $k \in [K]$,

$$R_{\mu\text{-supv}}(\mathbf{f}) - \mathbb{H}\left(\boldsymbol{\pi}\right)$$

$$\geq \underset{\substack{c^+,\{c_k^-\}_k, \\ \mathbf{x},\mathbf{x}^+,\{\mathbf{x}_k^-\}_k}}{\mathbb{E}} \left[ -\ln \frac{\sum_{k \in [K]} \exp(\mathbf{f}(\mathbf{x})^\top (\mathbf{f}(\mathbf{x}^+) - \mathbf{f}(\mathbf{x}_k^-)))}{K} \right] \tag{61}$$

$$\geq \underset{\substack{c^+,\{c_k^-\}_k, \\ \mathbf{x},\mathbf{x}^+,\{\mathbf{x}_k^-\}_k}}{\mathbb{E}} \left[ -\ln \frac{\exp(\mathbf{f}(\mathbf{x})^\top (\mathbf{f}(\mathbf{x}^+) - \mathbf{f}(\mathbf{x}^+))) + \sum_{k \in [K]} \exp(\mathbf{f}(\mathbf{x})^\top (\mathbf{f}(\mathbf{x}^+) - \mathbf{f}(\mathbf{x}_k^-)))}{K} \right]$$

$$\tag{62}$$

$$= \underset{\substack{c^+,\{c_k^-\}_k, \\ \mathbf{x},\mathbf{x}^+,\{\mathbf{x}_k^-\}_k}}{\mathbb{E}} \left[ \ln \frac{\exp(-\mathbf{f}(\mathbf{x})^\top \mathbf{f}(\mathbf{x}^+))}{\exp(-\mathbf{f}(\mathbf{x})^\top \mathbf{f}(\mathbf{x}^+)) + \sum_{k \in [K]} \exp(-\mathbf{f}(\mathbf{x})^\top \mathbf{f}(\mathbf{x}_k^-))} \right] + \ln K. \tag{63}$$

Finally, by using Lemma 5,

$$R_{\mu\text{-supv}}(\mathbf{f}) - \mathbb{H}\left(\boldsymbol{\pi}\right)$$

$$\geq \underset{c^+,\{c_k^-\}_k}{\mathbb{E}} \underset{\mathbf{x},\mathbf{x}^+,\{\mathbf{x}_k^-\}_k}{\mathbb{E}} \left[ -\ln \frac{\exp(\mathbf{f}(\mathbf{x})^\top \mathbf{f}(\mathbf{x}^+))}{\exp(\mathbf{f}(\mathbf{x})^\top \mathbf{f}(\mathbf{x}^+)) + \sum_{k \in [K]} \exp(\mathbf{f}(\mathbf{x})^\top \mathbf{f}(\mathbf{x}_k^-))} \right] + \ln K \tag{64}$$

$$- 2\ln\left\{ (K+1)\cosh(L^2) \right\} \tag{65}$$

$$= R_{\mathrm{cont}}(\mathbf{f}) + \ln K - 2\ln(K+1) - 2\ln\cosh(L^2), \tag{66}$$

which concludes the proof. $\qquad\square$

Unlike Theorem 1, Theorem 2 can only be extended to the coarse-grained $\mathcal{Y}$ from the case $\mathcal{Y} = [C]$. Indeed, the first expectation term on the right-hand side of Eq. (56) can be transformed as

$$\mathbb{E}_{c^+,\mathbf{x}} \left[ -\mathbf{f}(\mathbf{x})^\top \boldsymbol{\mu}_{c^+} + \sum_{c^- \in \mathcal{Y}} \pi_{c^-} \mathbf{f}(\mathbf{x})^\top \boldsymbol{\mu}_{c^-} \right] = \mathbb{E}_{c^+,\mathbf{x}} \left[ -\mathbf{f}(\mathbf{x})^\top \boldsymbol{\mu}_{c^+} + \sum_{c^- \in [C]} \pi_{c^-} \mathbf{f}(\mathbf{x})^\top \boldsymbol{\mu}_{c^-} \right], \quad (67)$$

because of the linearity of the expectation. On the other hand, it is not straightforward to lower-bound Eq. (56) by the expectation over the label set $[C]$ when $\mathcal{Y}$ is a strict subset of $[C]$.

## C  Essential Bounds of Mean Supervised and Contrastive Losses

This section provides a supplementary explanation of the essential lower bounds of the mean supervised and contrastive losses. The common approaches of CURL applies the normalization on representation, in order to employ the cosine similarity $\frac{\mathbf{f}(\mathbf{x})^\top \mathbf{f}(\mathbf{x}')}{\|\mathbf{f}(\mathbf{x})\|_2 \cdot \|\mathbf{f}(\mathbf{x}')\|_2}$ as the similarity metric. Then, it is reasonable to assume $\|\mathbf{f}(\mathbf{x})\|_2 \leq L$ for all $\mathbf{x}$ with our data representation $\mathbf{f}$. The normalized representation corresponds to the case $L = 1$.

When we introduce the constraint $\|\mathbf{f}(\mathbf{x})\|_2 \leq L$, the mean supervised loss and contrastive loss are restricted as well. As for the mean supervised loss,

$$R_{\mu\text{-supv}}(\mathbf{f}) \geq \inf_{\|\mathbf{f}'\|_2 \leq L} R_{\mu\text{-supv}}(\mathbf{f}') \quad (68)$$

$$= \inf_{\|\mathbf{f}'\|_2 \leq L} \mathbb{E} \left[ \ln \left( 1 + \sum_{c \neq y} \exp(\mathbf{f}'(\mathbf{x})^\top (\boldsymbol{\mu}_c - \boldsymbol{\mu}_y)) \right) \right] \quad (69)$$

$$= \ln \left( 1 + (C-1) \exp(-2L^2) \right) \quad (70)$$

$$(:= R_{\mu\text{-supv}}^*). \quad (71)$$

As for the contrastive loss,

$$R_{\text{cont}}(\mathbf{f}) \geq \inf_{\|\mathbf{f}'\|_2 \leq L} R_{\text{cont}}(\mathbf{f}') \quad (72)$$

$$= \inf_{\|\mathbf{f}'\|_2 \leq L} \mathbb{E}_{c^+,\{c_k^-\}, \mathbf{x} \, \mathbf{x}^+, \{\mathbf{x}_k^-\}} \left[ \ln \left( 1 + \sum_{k \in [K]} \exp(\mathbf{f}'(\mathbf{x})^\top (\mathbf{f}'(\mathbf{x}_k^-) - \mathbf{f}'(\mathbf{x}^+))) \right) \right] \quad (73)$$

$$\geq \inf_{\|\mathbf{f}'\|_2 \leq L} \mathbb{E}_{c^+,\{c_k^-\}, \mathbf{x}} \left[ \ln \left( 1 + \sum_{k \in [K]} \exp(\mathbf{f}'(\mathbf{x})^\top (\boldsymbol{\mu}_{c_k^-} - \boldsymbol{\mu}_{c^+})) \right) \right] \quad (74)$$

$$= \sum_{m=0}^{K} \binom{K}{m} \left(\frac{1}{C}\right)^m \left(1 - \frac{1}{C}\right)^{K-m} \ln\{1 + m + (K - m)\exp(-2L^2)\} \quad (75)$$

$$(:= R_{\text{cont}}^*), \quad (76)$$

where the Jensen's inequality is applied in the second inequality.

## D  Discussion of Existing Learning Bounds

In this section, we describe the existing learning bounds in details to make them comparable with our main results. Then, we further discuss the detailed comparison between our theory and existing works. Before the discussion, we need to introduce the *sub-class* loss (of the mean classifier), which is the supervised classification loss over a subset of classes:

$$R_{\text{sub}}(\mathbf{f}, T) := \mathbb{E}_{\mathbf{x},y} \left[ -\ln \frac{\exp(\boldsymbol{\mu}_y^\top \mathbf{f}(\mathbf{x}))}{\sum_{c \in T} \exp(\boldsymbol{\mu}_c^\top \mathbf{f}(\mathbf{x}))} \right], \quad (77)$$

where $T \subseteq [C]$ is a subset of classes and $y$ is drawn from the subset of $\boldsymbol{\pi}$ with respect to $T$.

**Arora et al.'s bound.** We introduce additional notation that Arora et al. (2019) use. For a subset of classes $T$,

- $Q \subseteq [C]$ is the set of distinct classes in $c^+, c_1^-, \ldots, c_K^-$
- $I^+ := \{k \in [K] \mid c_k^- = c^+\}$
- $\mathrm{Col} := \sum_{k \in [K]} \mathbb{1}_{\{c^+ = c_k^-\}} = |I^+|$
- $\rho_{\max}(T) := \max_{c \in T} \pi_c$
- $\rho_{\min}^+(T) := \min_{c \in T} \mathbb{P}_{c^+, \{c_k^-\}_k \sim \pi^{K+1}}(c^+ = c \mid Q = T, I^+ = \emptyset)$
- $\tau_K := \mathbb{P}(I^+ \neq \emptyset)$

Arora et al. (2019) prove a finite-sample learning bound in Theorem B.1. In its proof, Eq. (26) is a learning bound established for a fixed $\mathbf{f}$. For the comparison, we focus on their Eq. (26):

$$(1 - \tau_K) \underset{T \sim \pi^{K+1}}{\mathbb{E}} \left[ \frac{\rho_{\min}^+(T)}{\rho_{\max}(T)} R_{\mathrm{sub}}(\mathbf{f}, T) \right]$$
$$\leq R_{\mathrm{cont}}(\mathbf{f}) - \tau_K \underset{c^+, \{c_k^-\}_k \sim \pi^{K+1}}{\mathbb{E}} \left[ \ln(\mathrm{Col} + 1) \big| I^+ \neq \emptyset \right]. \quad (78)$$

We split the expectation term in the left-hand side as follows.

$$\mathbb{E}\left[ \frac{\rho_{\min}^+(T)}{\rho_{\max}(T)} R_{\mathrm{sub}}(\mathbf{f}, T) \right]$$
$$= \underbrace{\mathbb{P}(T \text{ covers } [C])}_{=v_{K+1}} \cdot \mathbb{E}\left[ \frac{\rho_{\min}^+(T)}{\rho_{\max}(T)} R_{\mathrm{sub}}(\mathbf{f}, T) \Big| T \text{ covers } [C] \right]$$
$$+ \mathbb{P}(T \text{ does not cover } [C]) \cdot \mathbb{E}\left[ \frac{\rho_{\min}^+(T)}{\rho_{\max}(T)} R_{\mathrm{sub}}(\mathbf{f}, T) \Big| T \text{ does not cover } [C] \right] \quad (79)$$
$$\geq v_{K+1} \frac{\rho_{\min}^+([C])}{\rho_{\max}([C])} \underbrace{R_{\mathrm{sub}}(\mathbf{f}, [C])}_{=R_{\mu\text{-supv}}(\mathbf{f})}. \quad (80)$$

Under the uniform class prior assumption ($\pi = 1/C \cdot \mathbf{1}$), $\rho_{\max}([C]) = 1/C$, and we can pick any class $c_0 \in [C]$ by the symmetry and $\rho_{\min}^+([C]) = \mathbb{P}(c^+ = c_0 \mid Q = [C], I^+ = \emptyset) = 1/C$. In addition,

$$\tau_K \underset{c^+, \{c_k^-\}_k \sim \pi^{K+1}}{\mathbb{E}} \left[ \ln(\mathrm{Col} + 1) \big| I^+ \neq \emptyset \right] = \mathbb{E}\left[ \ln(\mathrm{Col} + 1) \right] - (1 - \tau_K) \mathbb{E}\left[ \ln(\mathrm{Col} + 1) \big| I^+ = \emptyset \right]$$
$$(81)$$
$$= \mathbb{E}\left[ \ln(\mathrm{Col} + 1) \right]. \quad (82)$$

As a result, we obtain the following simplified expression in Table 1:

$$R_{\mu\text{-supv}}(\mathbf{f}) \leq \frac{1}{(1 - \tau_K)v_{K+1}} \left\{ R_{\mathrm{cont}}(\mathbf{f}) - \mathbb{E}[\ln(\mathrm{Col} + 1)] \right\}. \quad (83)$$

**Nozawa & Sato's bound.** The learning bound provided by Nozawa & Sato (2021, Theorem 8) involves a factor resulting from data augmentation and self-supervised learning setting. By dropping this (negative) factor, the learning bound is

$$R_{\mathrm{cont}}(\mathbf{f}) \geq \frac{1}{2} \left\{ v_{K+1} R_{\mu\text{-supv}}(\mathbf{f}) + (1 - v_{K+1}) \underset{T \sim \pi^{K+1}}{\mathbb{E}} [R_{\mathrm{sub}}(\mathbf{f}, T)] + \mathbb{E}[\ln(\mathrm{Col} + 1)] \right\} \quad (84)$$
$$\geq \frac{1}{2} \left\{ v_{K+1} R_{\mu\text{-supv}}(\mathbf{f}) + \mathbb{E}[\ln(\mathrm{Col} + 1)] \right\}, \quad (85)$$

resulting in the bound in Table 1. The sub-class loss may be safely dropped because it has the coefficient $1 - v_{K+1}$, which is expected to be exponentially small in $K$.

**Ash et al.'s bound.**   Ash et al. (2021, Theorem 5) provides the following learning bound

$$R_{\mu\text{-supv}}(\mathbf{f}) \leq \frac{2\left\lceil \frac{2(1-\pi_{(-1)})H_{C-1}}{K\pi_{(-1)}} \right\rceil}{(1-\pi_{(1)})^K} \left\{ R_{\text{cont}}(\mathbf{f}) - \tau_K \mathop{\mathbb{E}}_{c^+, \{c_k^-\}_k \sim \boldsymbol{\pi}^{K+1}} \left[ \ln(\text{Col}+1) \big| I^+ \neq \emptyset \right] \right\}. \tag{86}$$

By substituting $\boldsymbol{\pi} = 1/C \cdot \mathbf{1}$ and $\tau_K \mathbb{E}[\ln(\text{Col}+1) \mid I^+ \neq \emptyset] = \mathbb{E}[\ln(\text{Col}+1)]$, the bound in Table 1 is obtained.

**Detailed comparisons.**   As we stated in Section 4 of the main text, only our bound agrees well with the experimental fact that the larger $K$ is better for *all $K$ regions*:

- Arora et al. (2019): Large $K$ degrades the performance because of the label collision.
- Nozawa & Sato (2021): Large $K$ improves the performance for $K > C$.
- Ash et al. (2021): The optimal $K$ exists by the collision-coverage trade-off.

Even though the claim by Nozawa & Sato (2021) is similar with ours, we discovered a different underlying mechanism to support this idea, which leads to better explainability of empirical facts.

The proof of Nozawa & Sato (2021) is based on the idea of label coverage: The more negative samples we draw (larger $K$), the more likely the negative samples can cover all class labels. The upper bound based on this idea is only activated when $K > C$ because label coverage is impossible with $K \leq C$. This inability contradicts the real experiments including Chen et al. (2021), which showed that CURL exhibits reasonable performance even with small $K$.

Our proof leverages the idea that $R_{\text{cont}}$ and $R_{\mu\text{-supv}}$ have the similar log-sum-exp functional forms. This similarity casts $R_{\text{cont}}$ as a surrogate estimator of $R_{\mu\text{-supv}}$ and its estimation gap is reduced with larger $K$. Even with small $K$, the upper bound of $R_{\mu\text{-supv}}$ is loose but not vacuous thereby the estimation of $R_{\mu\text{-supv}}$ is still possible. Our theoretical claim reveals that the surrogate gap improves in $O(\ln 1/K)$ for all $K$ regions, which is in good agreement with the real experiments. Eventually, our theory provides practical feedback such that one may reduce $K$ (even smaller than $C$) to trade off the downstream performance with the computational cost.

# E   RELATIONSHIP TO MUTUAL INFORMATION (MI) ESTIMATION

The contrastive loss $R_{\text{cont}}$ we studied in this paper is also known as the InfoNCE loss (Oord et al., 2018), which is known to be deeply related to the multi-sample estimation of mutual information (MI) (Oord et al., 2018; Poole et al., 2019; Song & Ermon, 2020). Recently, the theoretical limitations of sample-based MI estimation have been analyzed (Gao et al., 2015; McAllester & Stratos, 2020). These studies revealed that a particular type of sample-based estimator of MI (Gao et al., 2015) or its lower bound (McAllester & Stratos, 2020) can be upper bounded by $O(\ln N)$ for the number of samples $N$. In this section, we discuss the implications of these limitations in the CURL setting.

Given two random variables $X$ and $Y$, suppose that we have $K + 1$ randomly drawn pairs $\{(x_i, y_i)\}_{i=1}^{K+1}$ from these random variables such that for all $(i, j)$, $(x_i, y_j)$ can be regarded as a positive pair when $i = j$, and otherwise can be regarded as a negative pair. Poole et al. (2019, Equation 10) derived the following lower bound for MI:

$$I(X; Y) \geq I_{\text{NCE}}^{K+1} := \mathbb{E}\left[ \frac{1}{K+1} \sum_{i=1}^{K+1} \ln \frac{\exp(s(x_i, y_i))}{\frac{1}{K+1} \sum_{j=1}^{K+1} \exp(s(x_i, y_j))} \right], \tag{87}$$

where $I(X; Y)$ is the MI between $X$ and $Y$, and $s(x, y)$ is a critic function. This lower bound estimator can be rewritten using $R_{\text{cont}}$ as follows:

$$I_{\text{NCE}}^{K+1} := \mathbb{E}\left[ \frac{1}{K+1} \sum_{i=1}^{K+1} \ln \frac{\exp(s(x_i, y_i))}{\frac{1}{K+1} \sum_{j=1}^{K+1} \exp(s(x_i, y_j))} \right] \tag{88}$$

$$= \mathbb{E}\left[ \frac{1}{K+1} \sum_{i=1}^{K+1} \ln \frac{\exp(s(x_i, y_i))}{\sum_{j=1}^{K+1} \exp(s(x_i, y_j))} \right] + \ln(K+1) \tag{89}$$

$$= \mathbb{E}\left[\frac{1}{K+1}\sum_{i=1}^{K+1}\ln\frac{\exp(s(x_i,y_i))}{\exp(s(x_i,y_i))+\sum_{j\neq i}\exp(s(x_i,y_j))}\right] + \ln(K+1) \qquad (90)$$

$$= \mathbb{E}\left[\ln\frac{\exp(s(x,x^+))}{\exp(s(x,x^+))+\sum_{j=1}^{K}\exp(s(x,x_j^-))}\right] + \ln(K+1) \qquad (91)$$

$$= -R_{\mathrm{cont}}(\mathbf{f}) + \ln(K+1). \qquad (92)$$

The first equality is obtained by putting the constant in the denominator outside. The third equality comes by replacing the notation $(x_i, y_i)$ with $(x, x^+)$ under the assumption that all $(x_i, y_i)$ come from the same iid distribution. By setting $s(x, y) := \mathbf{f}(x)^\top \mathbf{f}(y)$, we obtain the last equation.

Here, McAllester & Stratos (2020) gave the following theorem for the sample-based estimator of the lower bound on MI.

**Theorem 6** (McAllester & Stratos (2020) Theorem 1.1, informal). *Let $\widehat{I}^N$ be any mapping from $N$ samples of $(X, Y)$ to $\mathbb{R}$ that satisfies*

$$I(X;Y) \geq \widehat{I}^N(\{(x_i,y_i)\}_{i=1}^N) \qquad (93)$$

*in high probability, then the following relationship holds in high probability:*

$$\widehat{I}^N(\{(x_i,y_i)\}_{i=1}^N) \leq 2\ln N + 5. \qquad (94)$$

Since $I_{\mathrm{NCE}}^{K+1}$ satisfies the condition for $\widehat{I}^{K+1}$, we now have the following:

$$-R_{\mathrm{cont}}(\mathbf{f}) + \ln(K+1) \leq 2\ln(K+1) + 5 \implies R_{\mathrm{cont}}(\mathbf{f}) \geq -\ln(K+1) - 5. \qquad (95)$$

However, the right-hand statement always holds by the construction of $R_{\mathrm{cont}}$ for all $\mathbf{f}$ ($\forall \mathbf{f}, R_{\mathrm{cont}}(\mathbf{f}) \geq 0$). In other words, in the case of the CURL setting, McAllester & Stratos (2020)'s theorem does not restrict $R_{\mathrm{cont}}$, which means that the large $K$ effect investigated in our paper comes from a completely different mechanism from the above theorem. The existing studies on sample-based MI estimation are worthwhile in the sense that these works revealed the $O(\ln N)$ effect on the non-trivial estimators such as $k$-NN based estimator (Gao et al., 2015) or *any* kind of lower bound estimator (McAllester & Stratos, 2020).

## F    EXPERIMENTAL DETAILS

### F.1    SYNTHETIC DATASET

We used Adam (Kingma & Ba, 2015) optimizer with the weight decay of coefficient $0.01$ to all parameters. The mini-batch size was set to $B = 1\,024$ and the number of epochs was $300$. The learning rate was set to $0.01$ with `ReduceLROnPlateau` scheduler (patience: 10 epochs) provided by PyTorch (Paszke et al., 2019).

### F.2    CIFAR-10/100

We treated $10\%$ training samples as a validation dataset by sampling class uniformly. We used the original test dataset for testing. We used the same data-augmentation as in the CIFAR-10 experiment by Chen et al. (2020) during contrastive learning and linear supervised training of the linear classifier.

As a feature extractor $\mathbf{f}$, we modified the ResNet-18 (He et al., 2016) by following the convention of self-supervised representation learning (Chen et al., 2020, B.9); replacement of the first convolutional layer with a smaller one, removal of the first max-pooling layer, and replacement of the final fully-connected layer with a nonlinear projection head whose dimensional is $32$.[10]

Since we need to enlarge the negative samples size $K$ that depends on the size of mini-batches, we followed a large mini-batch training setting used in recent self-supervised learning (Chen et al., 2020;

---

[10]Unlike the reported results by Chen et al. (2021), smaller dimensionality, i.e., 32 gives better downstream accuracy on CIFAR-100 than 64 or 128. This difference might come from the differences in the loss function and positive pair's generation process.

Caron et al., 2020). We used LARC (You et al., 2017) optimizer wrapping the momentum SGD, whose momentum term was $0.9$. We applied weights decay of coefficient $10^{-4}$ to all parameters except for all bias terms and batch norm's parameters. The base learning rate was initialized at lr $\times \sqrt{B}$, where lr $\in \{2, 4, 6\} \times 1/64$ and mini-batch size $B = 1\,024$ inspired by SimCLR's squared learning rate scaling. As a learning rate scheduler for each iteration, we used linear warmup during the first 10 epochs and cosine annealing without restart (Loshchilov & Hutter, 2017) during the rest epochs. The number of epochs was $2\,000$.

We implemented our experimental code by using PyTorch (Paszke et al., 2019)'s distributed data-parallel training (Li et al., 2020) on 8 NVIDIA A100 GPUs provided by the internal cluster. Therefore we replaced the all batch normalization layer with `SyncBatchNorm` module provided by PyTorch.[11] To accelerate contrastive learning, we used automatic mixed-precision training provided by PyTorch.

### F.3 WIKI-3029

Wiki-3029 contains $3\,029$ English Wikipedia article pages. Each page consists of $200$ sentences. Since the dataset does not have the explicit train/validation/test splits, we split the dataset into $70\%/10\%/20\%$ train/validation/test datasets, respectively. As a pre-processing, we tokenized the dataset using torchtext's `basic_english` tokenizer. After tokenization, we removed the tokens whose frequency is less than $5$ in the training dataset. We did not use data augmentation.

We used fasttext (Joulin et al., 2017)'s based feature extractor.[12] In our preliminary experiments, only using a word embedding layer and average pooling among words perform better than either additional linear or nonlinear projection heads. A similar model to ours is also used in Ash et al. (2021). The dimensionality of the word embedding layer was $256$.

We mainly followed the same optimization setting as our CIFAR-10/100 experiments. We note that the mini-batch size $B = 2\,048$; the initial learning rate lr was selected in $\{1, 2, 3, 4\} \times 1/40$; no weights decay; the number of epochs was $90$; and perform linear warmup during the first 3 epochs. When we decrease $C$, the number of epochs is multiplied by $3\,000/C$ for simplicity.[13]

### F.4 CONTRASTIVE LEARNING

By following the data generation process in contrastive representation learning and existing work (Arora et al., 2019; Ash et al., 2021), we treated the supervised classes $\mathcal{Y}$ as latent classes $[C]$. After obtaining training/validation/test datasets as described above, we carefully constructed positive pairs for contrastive learning *before* training[14] as follows; We treated each sample in the training data as an anchor sample. We drew a different sample from the same latent class of each anchor sample as a positive sample in the training dataset. For negative samples, we drew $K$ negative samples from other samples in the same mini-batch by following the convention of self-supervised representation learning such as SimCLR (Chen et al., 2020). Since Chen et al. (2020) used all other samples as negative samples, the negative samples size and the size of mini-batches depend on each other: $K = 2B - 2$. To relax the effect of the difference of the mini-batch size when we change $K$, we drew $K$ samples without replacement from $2B - 2$ inspired by Ash et al. (2021). In this sampling, we guaranteed to draw at most one sample from each positive pair because we are concerned about the relation between the number of latent classes and $K$. We did not use validation and test datasets during contrastive representation learning.

### F.5 MEAN AND LINEAR CLASSIFIERS' EVALUATION

For evaluation, we reported the test accuracy values of mean and linear classifiers. For a linear classifier, we used Nesterov's momentum SGD, whose momentum coefficient was $0.9$ without weight

---

[11]See Wu & Johnson (2021, Sec. 6.2) for more detailed discussion of this replacement for contrastive learning.

[12]Arora et al. (2019) uses GRU-based feature encoder with frozen word embeddings of GloVe (Pennington et al., 2014) trained on commonCrawl.

[13]We found the contrastive learning did not yield good feature representations for a downstream task without this longer training.

[14]We can create the labeled dataset, especially with non-overlapped latent classes, if we draw positive samples at each iteration or epoch during optimization using stochastic gradient descent.

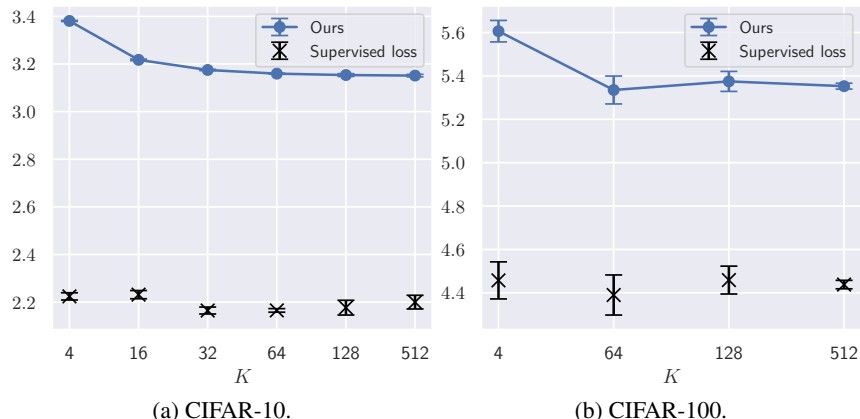

(a) CIFAR-10.       (b) CIFAR-100.

Figure 9: Enlarged Figure 1 for the detailed comparison between the proposed bound and the supervised loss on CIFAR-10/100 datasets. All value is an averaged value among three runs with a different random seed. Error bar indicates the standard deviation.

decay. We set the mini-batch size $B = 256$ and $B = 512$ for CIFAR-10/100 and Wiki-3029, respectively. We used cosine annealing without restart as a learning rate scheduler for each iteration. We set 100 and 30 epochs for CIFAR-10/100 and Wiki-3029 datasets, respectively. For CIFAR-10/100, we set learning rate as 0.03. For Wiki-3029, we searched the learning rate in $\{0.5, 1, 5, 10, 50\} \times 1/10^3$. The learning rate was scaled by using squared learning rate scaling. For linear evaluation of CIFAR-10/100, we used PyTorch's distributed data-parallel training. We calculated the test accuracy by using the best combination of the contrastive model and the hyper-parameter of a linear classifier that maximizes the validation accuracy. We repeated contrastive learning and downstream task's evaluation three times with different random seeds and reported the averaged values.

### F.6 Details of Figure 1

Before computing the upper bounds and supervised loss, we normalized feature representations $\mathbf{f}(\mathbf{x})$ learned in Appendix F.4 to ensure $L = 1$, which is the upper bound of $\|\mathbf{f}(\mathbf{x})\|_2, \forall \mathbf{x}$. For each random seed and the number of negative samples $K$, we selected learned feature encoder $\mathbf{f}$ that got the highest validation mean supervised accuracy in different learning rates of the optimizer of the contrastive learning. Then we calculated the test supervised loss value by using the selected contrastive models.

Using the same feature encoder with $L_2$ normalization, we calculated the contrastive loss on the test dataset. To do so, we created positive pairs by the same procedure on the test dataset as described in Appendix F.4. Negative samples were also drawn from the other samples in the mini-batches as the contrastive learning step described in Appendix F.4. To calculate the contrastive loss, we used the same batch size as the contrastive learning step and only one epoch. Since this contrastive loss calculation was stochastic due to the sampling of positive and negative samples, we repeated the contrastive loss calculation 25 times and averaged them to create plot Figure 1. Note that we used the theoretical values of $\tau_K, \upsilon_{K+1}, \mathbb{E} \ln(\mathrm{Col} + 1)$ that are shown in the existing upper bounds on Table 1 rather than the simulated values.

Figure 9 shows the enlarged version of Figure 1 and the same plot using CIFAR-100. This figure focuses on the detailed comparison between the test datasets' empirical supervised loss values and theoretical bounds. For both CIFAR-10/100 datasets, there were almost no changes in the supervised loss as $K$ varied, and the losses were slightly larger in the region where $K$ was small. These results are consistent with the theoretical estimation of the upper bounds (solid lines).

### F.7 Details of Figure 8

During minimization of the contrastive loss to learn $\mathbf{f}$ in Appendix F.4, we saved the model's weight at every 200 epochs. We reported the test mean supervised accuracy using $\mathbf{f}$ that maximized validation accuracy among different learning rate values.

### F.8    ADDITIONALLY USED LIBRARIES

In our experiments, we also used scikit-learn (Pedregosa et al., 2011) for train/val/test data splits. We created all plots by using matplotlib (Hunter, 2007) and seaborn (Waskom, 2021) via pandas (Reback et al., 2020) except for Figure 2. We managed our experiments' configuration using hydra (Yadan, 2019) and experimental results using Weights & Biases (Biewald, 2020). For effective parallelized execution of our experimental codes, we use GNU Parallel (Tange, 2021).

