# OpenReview forum: "Sharp Learning Bounds for Contrastive Unsupervised Representation Learning"
_ICLR.cc/2022/Conference — ICLR 2022 Submitted_

### Official Review · Reviewer_fm83 · 2021-10-31

**Correctness:** 2
**Technical Novelty And Significance:** 2
**Empirical Novelty And Significance:** 2
**Recommendation:** 6
**Confidence:** 5

**Main Review:**

This paper has made some interesting points, and the experiment results strongly support its theories. However, I do have a few outstanding concerns that prevent me from granting a positive recommendation. I am happy to discuss these during the author-reviewer interaction round, and I am more than willing to change my score if the author(s) have made proper changes or managed to convince me. Details of my review comments are given below.

**Strength**
* Strong theoretical analyses, clear writing, and nice illustrative figures. Overall the technical contents of the paper are easy to follow.
* Empirical results well aligned with the theories
* Very update-to-date coverage of the related literature, especially on the theoretical analyses of contrastive representation learning

**Weakness**
* My most concerning point is that: is this paper supervised representation learning in disguise? Because the contrastive loss used here clearly has leveraged class information (last equation on pp.2). This is clearly the setup in supervised contrastive learning [1]. In that case, it is not surprising that contrastive loss bounds the prediction loss: because they are essentially the same thing. In general, I do not think unsupervised learning can directly benefit supervised learning without some strong assumptions (see for example [2]).

* Apart from the above, another point I do not agree on is why larger K helps. The genuine unsupervised contrastive representation learning implicitly optimizes the MI across augmented views, and larger K drastically reduces estimator variance. The author(s) claimed that "the quantitative relationship between the negative sample size K and the MI estimation error has yet to be clearly known", which is not true, see theoretical analyses in [3,4].

* Following up on the last point, the author(s)' statement that "Note that Tian et al. (2020) and Tschannen et al. (2020) experimentally showed that maximizing MI does not necessarily lead to good representation." is also imprecise. To be more accurate, these works simply used tighter MI bounds for representation optimization, there is no guarantee that they indeed learned representations with higher MI. This view is supported by the empirical evidence from [5], where ground-truth representation MI strongly correlates with downstream classification, and tight MI bounds do not necessarily yield high MI representation.

* Unlike the author(s) have claimed, the gap between learning upper and lower bounds has nothing to do with the estimation variance of the mean supervised loss (pp. 4).

* Figure 4 is very compelling, my suggestion is to present it earlier in the text, which may better serve to prove the author(s) points.

* Per author(s)' theory, variance reduction is the key to performance gains. However, none of the experimental results shows the variance.

* What is the intuitive explanation for notation Col?

* Please label the equations with numbers.


**References**
* [1] P Khosla, et al. Supervised contrastive learning. NeurIPS 2020
* [2] J Robinson. Strength from weakness: Fast learning using weak supervision. ICML 2020
* [3] S Gao, et al. Efficient Estimation of Mutual Information for Strongly Dependent Variables. AISTATS 2015
* [4] D McAllester, et al. Formal Limitations on the Measurement of Mutual Information. AISTATS 2020
* [5] J Chen, et al. Simpler, Faster, Stronger: Breaking The log-K Curse On Contrastive Learners With FlatNCE. arXiv 2107.01152




**Summary Of The Paper:**

=== After Rebuttal ===

The author(s) have made many clarifications and improvements to the paper per reviewers' requests. I am happy to bump up my score by 1 to reflect the changes & promises made.

=================

This paper provides some new theoretical insights on how contrastive representation help to improve downstream classification performance. In particular, it fixes the mismatch between prior theoretical predictions and empirical observations, that a larger negative sample size leads to better downstream performance.

**Summary Of The Review:**

I am recommending weak rej at this stage. My main arguments are that (i) the author(s) actually analyzed supervised contrastive learning, not unsupervised contrastive learning; (2) need evidence to show the gains from large K are really from the variance reduction of sandwich bound, not from the variance reduction + bound tightening from the MI estimation perspective.

---

> ### Author Response · Authors · 2021-11-16
> **Reply to Reviewer fm83**
>
> We highly thank the reviewer for pointing out a lot of useful comments and literature from the professional perspective! Your comments must be addressed in future revisions. Please refer to the following answers to some of your questions and comments.
>
> ### Q1. Is this paper supervised representation learning in disguise?
> Please see the [general thread](https://openreview.net/forum?id=tDirSp3pczB&noteId=FKQDhtSehjam).
>
> ### Q2. There is a lack of reference to or misunderstanding related research on MI estimation.
> **A. On "the quantitative relationship between the negative sample size $K$ and the MI estimation error has yet to be clearly known":**
> This original phrase in the paper is unfortunately misleading. We intended to argue that the quantitative analysis of the bias-variance trade-off of the multi-sample MI estimators, which is the most closely related to contrastive loss, remains open. On the other hand, there are some quantitative analyses such as
> - the bias of general MI estimators (McAllester and Stratos (2020))
> - the variance of the NWJ/MINE estimators (Song et al. (2020))
>
> We revised to clarify this point and mention important literature such as McAllster and Stratos (2020) in the updated manuscript.
>
> **B. On why larger $K$ helps:**
> While the reviewer's argument that CURL implicitly optimizes the MI across augmented views makes sense, our focus is on why CURL improves downstream classification performance, which is a different concept from MI. From this viewpoint, connecting the contrastive loss and the supervised loss seems reasonable. Eventually, we think our finding is reasonable such that the estimation gap of the mean supervised loss shrinks with the larger $K$. It must be another interesting way to theoretically study the relationship between MI and downstream performance.
>
> We suppose that theoretical investigations of the relationship between MI and downstream performance must be interesting. Yet, the current MI theory still has a hurdle to overcome. As seen in McAllester and Stratos (2020), any variational MI estimators admit the $(O(\ln N))$ upper bound ($N$: # of samples), which supports many qualitative arguments such that the multi-sample MI estimators (e.g. InfoNCE) have high bias but low variance (Song et al. (2020)). Here, the InfoNCE loss can be connected to $R\_{\mathrm{cont}}$ by Poole et al. (2019) such that $I(X, Y) \ge I_{\mathrm{NCE}}^{K+1} := - R\_{\mathrm{cont}} + \ln (K +1)$, where $K + 1$ is the number of samples (one positive plus $K$ negative samples). From this relationship, it is often pointed out that InfoNCE loss, or contrastive loss, maximizes the MI $I(X, Y)$ between augmented views $X$ and $Y$. By invoking the $(O(\ln N))$ upper bound, we obtain a trivial bound $R\_{\mathrm{cont}} \ge 0$, meaning that the bias-variance trade-off of the MI estimation does not necessarily lead to the trade-off in downstream classification. Detailed discussions have been added in Appendix E.
>
> Hence, we currently focus on the direct relationship between the contrastive loss and (mean) supervised loss.
>
> **C. On Tian et al. (2020) and Tschannen et al. (2020):**
> Thank you for pointing out the precise statement. We will modify the manuscript as suggested in future revisions.
>
> ### Q3. Unlike the author(s) have claimed, the gap between learning upper and lower bounds has nothing to do with the estimation variance of the mean supervised loss.
> As mentioned in footnote 6, the gap between upper and lower bounds is not the statistical "variance" in a rigorous sense. Since we intend to argue that an increase in $K$ narrows the range of possible values of the loss and reduces the variability, we will replace the "estimation variance" with "estimation gap" in the further revision.
>
> ### Q4. None of the experiments shows the "variance" of the loss even the authors claimed the variance reduction is the key to performance gains.
> As we discussed above, the term "variance" was misleading, and we are considering rephrasing it. Also, we did not intend to say the variance reduction is the key to performance gain. Rather, our main claim is that the $R\_{\mathrm{cont}}$ essentially behaves as the surrogate estimate of $R\_{\mu-\mathrm{supv}}$. These two metrics correlate, and the estimation gap reduces with an increase in $K$. Empirically, this gap reduction resulted in the variance reduction of the mean supervised loss in Figure 6 (right).
>
> ### Q5. What is the intuitive explanation for notation $\mathrm{Col}$?
> $\mathrm{Col}$ is the number of negative samples whose latent class is the same as the anchor sample's latent class. So $\mathrm{Col}$ increases when $K$ increases.
>
> ### Q6. Suggestions on the manuscript: change the place of Figure 4 and label the equations with numbers.
> Thank you for your concrete suggestion to improve our manuscript. As in the latest manuscript, we've labeled all the equations for ease of discussion.  We will modify the place of Figure 4 in the further revision.
>
> (Reference follows)

---

> > ### Author Response · Authors · 2021-11-16
> > **Missing reference**
> >
> > ### Reference
> >
> > - D. McAllester, et al. Formal Limitations on the Measurement of Mutual Information. In _AISTATS_, 2020.
> > - B. Poole, et al. On Variational Bounds of Mutual Information. In _ICML_, 2019.
> > - Y. Tian, et al. What Makes for Good Views for Contrastive Learning? In _NeurIPS_, 2020.
> > - M. Tschannen, et al. On Mutual Information Maximization for Representation Learning. In _ICLR_, 2020.
> > - J. Song, et al. Understanding the Limitations of Variational Mutual Information Estimators. In _ICLR_, 2020.

---

### Official Review · Reviewer_unTX · 2021-10-31

**Correctness:** 4
**Technical Novelty And Significance:** 3
**Empirical Novelty And Significance:** 3
**Recommendation:** 8
**Confidence:** 4

**Main Review:**

Strengths:
1.	The proposed upper-bound on both contrastive loss and downstream supervised loss is lower compared to other upper-bounds, which make it sharper.
2.	The upper-bound on supervised loss decreases when K (the number of negative examples) increases. This does not happen for other upper-bounds and is important for understanding why the performance of models gets better with more negative samples.
3.	The empirical experiments on the synthetic datasets, CIFAR-10, and CIFAR-100 further show the clear advantage of this upperbound compared to the others.

Weaknesses:
1.	This upper-bound does not include the interaction between K and C (the number of classes). This makes it monotonically decrease when K increases, which is contradict to the empirical results on CIFAR-10 or CIFAR-100 (though the loss values do not change that much, the performance indeed decreases instead of increases as predicted). This interaction is in fact shown in earlier upper-bound. I wonder how this interaction can also be introduced into the current upperbound, especially giving the intuition that with more negative samples, there will also be more examples in the same class treated as positive examples.
2.	The proposed upperbound on downstream supervised loss is based on the intermediate upperbound on the mean supervised loss, which essentially uses the low-dimensional embeddings as the final representations to do the classification. Although this would serve as a right intermediate upperbound when the embedding is only one linear layer away from the typical layer the category readout is extracted, this would not be true when the embedding is several layers away from the typical layer, which is actually the case in most SOTA algorithms now. Can the authors explain how this would influence the results in this work?


**Summary Of The Paper:**

This work proposes a new theoretical upper-bound for contrastive unsupervised learning loss and its downstream supervised loss. Compared to previously published upper-bounds, the newly proposed one is lower and decreases when the number of negative samples increases, which is expected according to empirical observations, but not shown in previous upper-bounds. The authors use experiments on multiple datasets to verify the theoretic results and the gaps between this and earlier upper-bounds.

**Summary Of The Review:**

This work proposes a sharp loss upperbound for contrastive learning and its downstream supervised learning. It is validated by the empirical experiments conducted by the authors on different datasets, but lacks the interaction between the number of classes and the number of negative samples. Because it’s indeed lower than the earlier upperbound, I think it would be useful for the community and would recommend for acceptance.

---

> ### Author Response · Authors · 2021-11-16
> **Reply to Reviewer unTX**
>
> We appreciate your attention and feedback on our technical details. If the following initial answers do not capture what you meant, we are very happy to clarify further.
>
> ### Q1. Can this theory also incorporate the interactions between $C$ and $K$ that exist in the existing theories?
> Suppose that you meant the label collision-coverage trade-off in the existing theories by "interaction between $C$ and $K$". That is, the CURL performance initially becomes better with the better class coverage but gradually degenerates because of the positive class collision among the sampled classes, as we increase $K$.
>
> In the existing bounds, there does exist such a trade-off in contrast to our results, as you pointed out. The term $\ln(\mathrm{Col}+1)$ in every bound in Table 1 represents the label collision effect. However, we consider this trade-off is only an artifact needed to derive the learning bound in the existing works. We introduced a new technique to show the "correlation" between the contrastive loss and mean supervised loss directly, and successfully eliminated this trade-off. Since this is the best upper bound so far as we see numerically in Figures 3 and 4, we think the trade-off is not essential at this moment.
>
> ### Q2. Is there any problem in treating mean supervised loss as an intermediate bound for models with multiple embedding layers, as is the case with recent SOTA models?
> We presume that you are asking if the analysis based on the mean supervised loss can deal with e.g. SimCLR, which in practice removes the last few layers to obtain representation.
>
> The answer is no. This kind of training trick has yet to be handled in any theory so far and remains open to be investigated. Nevertheless, we believe this does not undermine our contributions because we are strongly motivated to elucidate how $K$ affects contrastive learning and it is known that the mean supervised analysis empirically correlates with how SimCLR behaves (Nozawa & Sato, 2021), precisely, "$\mu$ acc" and "Linear acc w/o" on Table 1.

---

> > ### Comment · Reviewer_unTX · 2021-11-30
> > **response**
> >
> > I have read the rebuttal and I am happy to keep my current score. I look forward to future works based on this result!

---

### Official Review · Reviewer_d3WX · 2021-11-02

**Correctness:** 3
**Technical Novelty And Significance:** 1
**Empirical Novelty And Significance:** Not applicable
**Recommendation:** 6
**Confidence:** 4

**Main Review:**

Strengths:
(1)	Its theoretical results could explain larger negative samples improve the classification performance in contrastive learning by establishing lower and upper bound of downstream classification loss.
(2)	The authors also do some experiments to investigate their theories.

Weaknesses:
(1)	My biggest concern is that this work does not provide any useful and new insights. It can only explains a well-known experimental fact that larger negative samples improve the classification performance, and does not provide any insights to further improve contrastive learning which is actually the key for theoretical work in my viewpoint. for a theoretical work, providing a more sharp bound is not the goal but providing new insights is. For example, Arora first establish the relation between pretext contrastive loss and the downstream classification loss and shows why contrastive loss can work well for downstream tasks.  However, in this work, I did not find such kinds of insights.

(2)	the bound in Nozawa & Sato actually also partially support that larger negative samples improve the classification performance, which can be illustrated by Fig. 3 (a). The authors may claim for possible best R*cont, the bound in Nozawa & Sato cannot explain the experimental fact. Actually, R*cont  is obtained under a simplified case where data augmentation is not considered, and the classification head in f is not discarded when training on downstream tasks, etc. So R*cont may not be achievable for one real contrastive learning or apart from the real optimal contrastive loss since the simplified case are different from the real contrastive learning.

(3)	The authors should discuss the novelty of the proof techniques compared with existing one, such as Arora et al, Nozawa & Sato. It seems that the authors use the property that the function f is bounded to give a tight bound. So it is better to discuss your technique differences.


**Summary Of The Paper:**

This work establishes a downstream classification loss bound for contrastive learning which shows that larger negative samples improve the classification performance. Existing works cannot explain the experimental fact that larger negative samples improve the classification performance.

**Summary Of The Review:**

Overall, although this work provides new tighter bound for contrastive learning, it does not provide new insights and new proof technique/frameworks.

---

> ### Author Response · Authors · 2021-11-16
> **Reply to Reviewer d3WX**
>
> We appreciate the reviewer's valuable questions on our manuscript! We hope the following answers resolve your concerns.
>
> ### Q1. What are the useful or new insights that can be gained from this theory?
> As acknowledged by the reviewer URqh, we believe our theory potentially provides practical feedback to the experiments. It is supported by the following two points.
>
> **A. Our theory explains the underlying mechanism of the CURL behavior more accurately than the existing works.**
> The claims of the existing CURL theories can be roughly summarized as follows.
>
> - Arora et al.: large $K$ degrades the performance by label collision
> - Nozawa and Sato: large $K$ improves the performance by label coverage
> - Ash et al.: the optimal $K$ exists by the collision-coverage trade-off
>
> Why do they contradict each other? It is because their discussions were based on the upper bounds not sharp enough. We could draw arbitrary conclusions from vacuous bounds at worst. By contrast, we provided the sharper upper bound and its sharpness can be numerically and empirically observed in Figures 3 and 4. In addition, the upper and lower bounds have the same intercept order $O(\log(1/K))$. All of them support our claim "larger $K$ helps to reduce the estimation gap" and resolve the controversy so far. The difference between ours and Nozawa & Sato (2021) is discussed in Q2 later.
>
> **B. Sharper bounds provide better estimates of the necessary negative samples.**
> Our upper bound suggests the qualitatively same argument as Nozawa & Sato (2021), "the larger $K$ is better." However, our upper bound does not only provide sharper estimates of the mean supervised loss $R\_{\mu-\mathrm{supv}}$ but also explain $R\_{\mu-\mathrm{supv}}$ can be estimated when $K \le C$. The difference between the two works is detailed in Q2. Eventually, our theory provides practical feedback such that one may reduce $K$ (even smaller than $C$) to trade off the downstream performance with the computational cost. In addition, our upper and lower bounds can be combined to roughly estimate the possible range of the mean supervised loss from the current contrastive loss alone *before* fine-tuning. We believe these insights can accelerate experimental research.
>
> ### Q2. What is the contribution/difference of this research to Nozawa & Sato (2021), which has already shown that large $K$ improves performance?
>
> Even though one of our main claims (larger $K$ is better) is the same as Nozawa & Sato (2021), we discovered a different underlying mechanism to support this idea.
>
> The proof of Nozawa & Sato (2021) is based on the idea of label coverage: The more negative samples we draw (larger $K$), the more likely the negative samples can cover all class labels. The upper bound based on this idea is only activated when $K > C$ because label coverage is impossible with $K \le C$. This inability contradicts the real experiments provided by e.g. Chen et al. (2021), which showed that CURL exhibits reasonable performance even with small $K$.
>
> Our proof leverages the idea that $R\_{\mathrm{cont}}$ and $R\_{\mu-\mathrm{supv}}$ has the similar log-sum-exp functional forms. This similarity cast $R\_{\mathrm{cont}}$ as a surrogate estimator of $R\_{\mu-\mathrm{supv}}$ and its estimation variability is reduced with larger $K$. Even with small $K$, the upper bound of $R\_{\mu-\mathrm{supv}}$ is loose but not vacuous thereby the estimation of $R\_{\mu-\mathrm{supv}}$ is still possible. Our theoretical claim reveals that the surrogate gap improves in $O(\log(1/K))$ for *all $K$ regions*, which is in good agreement with the real experiments.
>
> Chen et al. Intriguing Properties of Contrastive Losses. In _NeurIPS_, 2021.
>
> ### Q3. What is the novelty of the proof techniques of this study, especially about the bounded property of the function $f$?
> Please see the [general thread](https://openreview.net/forum?id=tDirSp3pczB&noteId=FKQDhtSehjam).

---

> > ### Comment · Reviewer_d3WX · 2021-11-30
> > **response**
> >
> > Thanks authors' feedback. It partially addresses my concerns, especially for Q2. So I raise my score. For new insights, I still think that one theoretical work should not only provide a tight bound but provide solutions to further improve the method.

---

### Official Review · Reviewer_URqh · 2021-11-03

**Correctness:** 4
**Technical Novelty And Significance:** 3
**Empirical Novelty And Significance:** 3
**Recommendation:** 8
**Confidence:** 3

**Main Review:**

Strength:

* The new upper bound is tighter than the existing works, and provides a better insights of having the large size of negative example that people found useful in practice.  One thing I enjoy during reading is Section 3. After having lower bound, the discussion of feasible region is interesting, and also provide some aspects of having "small K", which is also quite aligned with some empirical findings.

* The numerical figures in Sec 4 is a great illustration to understand and compare different bounds.

Suggestions:

* It would be great to extend the mean classifier to a more general linear classifier setting, which is more commonly used.

* Currently, the success of many contrastive learning heavily relying on data augmentations. It would be great the augmentation can be formulated into the analysis (e.g. as some stochastic process .. etc)

**Summary Of The Paper:**

The paper studies the learning bounds of popular representation learning. Not only the improved tighter upper bound is provided, the lower bound is also studied. The paper is well written with thorough discussion and nice example for illustration.

**Summary Of The Review:**

The paper is well written with many great figures to explain ideas. The improved upper bound and lower bound are studied, which I think it's good to the community.  However, I don't closely follow the theory development of representation learning, which I can only follow the derivations in the paper, and judge from a practitioner perspective.

---

> ### Author Response · Authors · 2021-11-16
> **Reply to Reviewer URqh**
>
> We appreciate you for recognizing our paper's contributions. The answers to your concerns (extension for a linear classifier & data augmentation) have been addressed in the [general thread](https://openreview.net/forum?id=tDirSp3pczB&noteId=FKQDhtSehjam). We are happy to answer any further questions.

---

### Official Review · Reviewer_Dmou · 2021-11-04

**Correctness:** 3
**Technical Novelty And Significance:** 3
**Empirical Novelty And Significance:** 1
**Recommendation:** 6
**Confidence:** 4

**Main Review:**

The paper address an important theoretical problem in contrastive learning. The paper is well-written and the contributions are stated clearly. I have a few questions and concerns.

1. Is CURL really unsupervised? The paper claims to analyze the “unsupervised” representation learning, however, the positive samples are drawn from class-conditioned distribution, which requires ground truth labels to estimate. I know this step is necessary to have some closed-form analyses, but this makes the study a bit deviated from the recent contrastive learning approaches. Does the observation still holds for general unsupervised contrastive learning, where positive samples are drawn via data augmentations? At least, in the experimental section, it would be great to see some results with standard framework such as SimCLR. Even though the results might not perfectly match the theory, some discussions are still valuable.


2. The class sampling assumption is stronger than Arora et al. [1]. In particular, the latent classes used in [1] is not equivalent to the output classes for supervised classification task. Can the current framework generalizes to this scenario?


3. Theorem 1 and 2 are nice. Maybe the authors can provide some proof sketch or at least discuss the key tool use to tighten the bounds? It seems that most of the approaches including Thm 1 applying Jensen Ineq at first, what makes the bound tighter? Giving a high level intuition of proof technique can help reader understand the theorems better.

4. Are the bounds also tighter in term of generalization error? With access to only finite samples, the bound could behave different from its expected value.

Overall, the theorems are solid and sound. However, I am concern about the applicability of the bounds. The assumption makes it hard to quantify the behavior of “actual” unsupervised contrastive learning algorithms. The insights from the analyses seem to not directly applicable to practical setting.



[1] Arora et al., A Theoretical Analysis of Contrastive Unsupervised Representation Learning, 2019

**Summary Of The Paper:**

The paper analyzes contrastive learning by bounding the supervised loss with contrastive loss. The bounds are tighter than those proposed by previous works.

**Summary Of The Review:**

The paper is well-written, but I am concern about the applicability of the bounds due to its assumptions. The authors claim to analyze an unsupervised algorithm, but the assumptions do require the access to full supervision.

---

> ### Author Response · Authors · 2021-11-16
> **Reply to Reviewer Dmou**
>
> We appreciate the reviewer's valuable questions on our manuscript! We hope the following answers resolve your concerns.
>
> ### Q1 Is CURL really unsupervised? Does the observation still hold for general unsupervised contrastive learning, where positive samples are drawn via data augmentations?
> Please see the [general thread](https://openreview.net/forum?id=tDirSp3pczB&noteId=FKQDhtSehjam).
>
> ### Q2. Can the current framework generalize to Arora et al.'s class assumption?
>
> As noted in footnote 4, Theorem 1 (the CURL upper bound) can deal with the following cases:
>
> 1. $\mathcal{Y} \subset [C]$
> 2. $\mathcal{Y}$ is a coarse-grained set of $[C]$.
>
> More concretely, we discuss it in the last paragraph of the proof of Theorem 1 in Appendix B.
>
> On the other hand, Theorem 2 (the CURL lower bound) can only deal with the coarse-grained case. It remains open, while we believe that the results are still valuable as a first step to discuss the lower bound.
>
> ### Q3. Maybe the authors can provide some proof sketch or at least discuss the key tool used to tighten the bounds? It seems that most of the approaches including Theorem 1 applying Jensen inequality at first, what makes the bound tighter?
> Please see the [general thread](https://openreview.net/forum?id=tDirSp3pczB&noteId=FKQDhtSehjam).
>
> ### Q4. Are the bounds also tighter in terms of generalization error? With access to only finite samples, the bound could behave differently from its expected value.
>
> We mentioned it at the end of Section 3.1:
>
> > By applying either the high-probability bound (Arora et al., 2019) or PAC-Bayesian analysis (Nozawa et al., 2020), Theorem 1 (Theorem 2 as well) can be naturally extended to the form $R_{\mu\mathrm{supv}}(\hat{f}) ≤ R_{\mathrm{cont}}(f) + \Delta_\mathrm{U} + \chi$ with a complexity term $\chi$, where $\hat f$ is the empirical minimizer of the contrastive loss. Since this is a routine, we omit the high-probability bounds.
>
> It is true that "the bound could behave differently from its expected value," while we are primarily interested in how the contrastive loss behaves under a sufficiently large dataset. If the contrastive loss does not behave with the large sample limit as expected, it is almost hopeless under the finite sample regime.

---

> > ### Comment · Reviewer_Dmou · 2021-11-25
> > **Response**
> >
> > Thank you for the clarifications. I understand the same theoretical framework is adopted by some previous works. Unfortunately, the bar has been set higher in research about contrastive learning theory. There are some works, e.g., HaoChen et al., 2021, start considering frameworks that are closer to practical setting, which could provide more useful insights. Nevertheless, the proposed analysis is still valuable and I will keep my score unchaned.

---

### Author Response · Authors · 2021-11-16
**General concerns (responding multiple reviewers)**

We would like to highly appreciate all reviewers for their constructive feedback and comments. This thread is devoted to dealing with questions and concerns raised by multiple reviewers.

### 1. Is the problem setting considered in this paper really unsupervised? (Dmou, URqh, fm83)
As the reviewers mentioned, we assume the supervised classes and conditional independence, which may seem a bit awkward as unsupervised learning. However, we think they are not a big deal.

As for the conditional independence assumption (data are drawn from the class-conditional distribution), we believe that considering the class-conditional distribution is mild since the classes are just latent classes, not supervised classes. This is a common approach to analyze contrastive unsupervised representation learning in light of downstream classification (Arora et al., 2019).

As for the supervised classes (we are given the same supervised classes across both the training and test phases), this is still an acceptable formulation for two reasons.

1. Our learning (upper/lower) bounds can deal with the case where the test classes (supervised label set $\mathcal{Y}$) is a coarse-grained set of the training classes (latent classes $[C]$) under this formulation (see footnote 4). Since representation learning generally aims to elicit as many underlying concepts as possible in training and help test prediction, this coarse-grained setup is somewhat natural.
2. It is essential to assume data are generated from class-conditional distributions (like Arora et al. (2019) and the follow-up works) in order to establish some meaningful results on downstream classification.

### 2. Extension from the mean classifier to more general classifiers (URqh, unTX)
Indeed, our theoretical analysis on the upper bound holds with general linear classifier as explained in the paragraph `Evaluation of representations` on page 3,

> If the mean supervised loss is successfully bounded from above, we end up a bound on the supervised loss through $ \inf\_{\mathbf{W} \in \mathbb{R}^{C \times h}} R\_{\mathrm{supv}}(\mathbf{W} f) \leq R\_{\mathrm{\mu-supv}}(f)$. For this reason, an upper bound on $R\_{\mathrm{\mu-supv}}$ is an intermediate milestone that we seek throughout this paper.

For lower bound analysis, since the correlation between the loss of any general linear classifier and $R_{\mathrm{cont}}$ is non-trivial, we provided the lower bound of $R_{\mu-\mathrm{supv}}$ as a representative target.

### 3. Extension to incorporate data augmentation (URqh, Dmou, fm83, d3WX)
Nozawa & Sato (2021) has already analyzed the self-supervised setting and revealed that the learning bound incurs an excessive factor (see $d(f)$ in their equation 7). This excessive factor can be readily incorporated into our bounds as well, and our main technical findings would remain the same. For finer analysis with data augmentation, we believe that additional assumptions like the expansion assumption (Wei et al., 2021) are needed. We did not provide our results with data augmentation to avoid the digression and focus on the sharpness of the CURL bounds at least in the current work (see footnote 3 as well).

Wei et al. Theoretical Analysis of Self-Training with Deep Networks on Unlabeled Data. In _ICLR_, 2021.


### 4. Proof sketch and intuition of why the bound becomes better (Dmou, d3WX)
Loosely speaking, our technique leverages the intuition that the contrastive loss and mean supervised loss "correlate with each other" (both are some kinds of the log-sum-exp function), while the existing works including Arora et al. (2019) approximate the mean supervised loss with the contrastive loss by taking the expectation over latent classes. This expectation results in an exponentially small coefficient in the bound, in contrast to our approach. The variability of this correlation depends on the number of the latent class $C$, the negative sample size $K$, and the maximum norm of the learned representation $L$. In particular, to restrict the size of the norm $L$, we assumed a bounded property for the function $f$. Our analysis is novel in that it focuses on the correlation between the two kinds of losses, leading to a sharp bound. We will discuss it in the future revision.

---

### Author Response · Authors · 2021-11-20
**Notification of revised manuscript**

We appreciate all the reviewers for the constructive and insightful discussions and comments on our paper. According to the reviewers’ discussions and suggestions, we have updated the manuscript from its submitted version as follows:
- numbered all the equations for ease of discussion
- rephrased the misleading word "estimator variance" to "estimator gap"
- refined the description of existing studies related to mutual information (MI) maximization  (Section 5)
- added an intuitive explanation of our proofs (Section 3.1)
- moved the empirical comparison of theoretical bounds on an earlier page to emphasize our contribution (page 2)
- added the detailed discussion on the relationship to MI estimation (Appendix E)
- added the detailed discussion on the comparison with existing works and the insights (Appendix D)

All modifications listed above are highlighted in blue color.

---

### Decision · Program_Chairs · 2022-01-20

**Decision:**

Reject

**Comment:**

The paper presents an analysis of the benefit of unsupervised contrastive learning for downstream classification tasks using the cross-entropy loss. Building on prior work, the authors show that the contrastive loss can be bounded in terms of the cross=entropy term and an “intercept” term which depends logarithmically on the number of negative samples per positive sample (for contrastive learning) rather than polynomially as in the prior work.

There are several differences between the setting here and that of the prior work by Arora et al. (2019). First, the work here focuses only on cross-entropy loss and leverages the similarity of the loss structure between the contrastive loss and the cross-entropy loss. Second, the assumptions here are different, e.g., boundedness of the representation. Finally, the assumption that latent classes are the same as the label classes (which is not the case in the prior work) is significantly restrictive.

The writing is poor and the presentation is not clear. Despite the title and various references to learning bounds in the abstract and the main text, there are no learning bounds in the paper. The main result is to bound the contrastive loss in terms of the cross-entropy loss under the assumption that the latent classes and the label classes coincide. Authors state that getting generalization bounds is routine and, therefore, they chose not to give them — I do not see how generalization bounds follow in a straightforward manner here, and even if they do, it is important to write them for completeness.

The main contribution here is that the bounds depend logarithmically in K — the number of negative samples per positive sample — compared to sqrt{K} in the previous work. The previous bound however holds for Lipschitz losses as well, for e.g., hinge loss. So the question remains whether this improvement is only for the cross-entropy loss. Regardless, K is typically small in practical applications. Even the experiments in the paper (Figure 7) suggest that the performance degrades for larger K even on simple tasks. So, the improvement is really somewhat insignificant.

The reviewers were generally positive and appreciated the paper. However, in the light of comments above (of which I am quite certain), unfortunately, I am unable to accept the paper at this point. I believe the comments above (and from the other reviewers) will help improve the overall quality of the paper. I encourage the authors to incorporate the feedback and work towards a stronger submission.